

# Urban influence on the concentration and composition of submicron particulate matter in central Amazonia

Suzane S. de Sá (1), Brett B. Palm (2), Pedro Campuzano-Jost (2), Douglas A. Day (2), Weiwei Hu (2), Gabriel Isaacman-VanWertz[a] (3), Lindsay D. Yee (3), Joel Brito[b] (4), Samara Carbone[c] (4), Igor O. Ribeiro (5), Glauber G. Cirino[d] (6), Yingjun J. Liu[e] (1), Ryan Thalman[f] (7), Arthur Sedlacek (7), Aaron Funk (8), Courtney Schumacher (8), John E. Shilling (9), Johannes Schneider (10), Paulo Artaxo (4), Allen H. Goldstein (3), Rodrigo A.F. Souza (5), Jian Wang (7), Karena A. McKinney[g] (1), Henrique Barbosa (4), M. Lizabeth Alexander (11), Jose L. Jimenez (2), Scot T. Martin[*] (1, 12)

(1) School of Engineering and Applied Sciences, Harvard University, Cambridge, Massachusetts, USA
(2) Department of Chemistry and Cooperative Institute for Research in Environmental Sciences, University of Colorado, Boulder, Colorado, USA
(3) Department of Environmental Science, Policy, and Management, University of California, Berkeley, California, USA
(4) Institute of Physics, University of São Paulo, São Paulo, Brazil
(5) School of Technology, Amazonas State University, Manaus, Amazonas, Brazil
(6) National Institute for Amazonian Research, Manaus, Amazonas, Brazil
(7) Brookhaven National Laboratory, Upton, New York, USA
(8) Department of Atmospheric Sciences, Texas A&M University, College Station, Texas, USA
(9) Atmospheric Sciences and Global Change Division, Pacific Northwest National Laboratory, Richland, WA, USA
(10) Particle Chemistry Department, Max Planck Institute for Chemistry, Mainz, Germany
(11) Environmental Molecular Sciences Laboratory, Pacific Northwest National Laboratory, Richland, Washington, USA
(12) Department of Earth and Planetary Sciences, Harvard University, Cambridge, Massachusetts, USA
[a] Now at Department of Civil and Environmental Engineering, Virginia Tech, Blacksburg, Virginia, USA
[b] Now at Laboratory for Meteorological Physics (LaMP), University Blaise Pascal, Aubière, France
[c] Now at Federal University of Uberlândia, Uberlândia, Minas Gerais, Brazil
[d] Now at Department of Meteorology, Geosciences Institute, Federal University of Pará, Belém, Brazil
[e] Now at University of California, Berkeley, California, USA
[f] Now at Department of Chemistry, Snow College, Richfield, Utah, USA
[g] Now at Colby College, Waterville, Maine, USA

[*]To Whom Correspondence Should be Addressed

*E-mail: scot_martin@harvard.edu*

*https://martin.seas.harvard.edu/*





**Abstract**
Fundamental to quantifying the influence of human activities on climate and air quality is
an understanding of how anthropogenic emissions affect the concentrations and composition of
airborne particulate matter (PM). The central Amazon basin, especially around the city of
Manaus, Brazil, has experienced rapid changes in the past decades due to ongoing urbanization.
Herein, changes in the concentration and composition of submicron PM due to pollution
downwind of the Manaus metropolitan region are reported as part of the GoAmazon2014/5
experiment. A high-resolution time-of-flight aerosol mass spectrometer (HR-ToF-AMS) and a
suite of other gas- and particle-phase instruments were deployed at the "T3" research site, 70 km
downwind of Manaus, during the wet season. At this site, organic components represented on
average $79 \pm 7\%$ of the non-refractory $PM_1$ mass concentration, which was in the same range as
several upwind sites. The organic $PM_1$ was, however, considerably more oxidized at T3
compared to upwind measurements. Positive-matrix factorization (PMF) was applied to the time
series of organic mass spectra collected at the T3 site, yielding three factors representing
secondary processes ($73 \pm 15\%$ of total organic mass concentration) and three factors
representing primary anthropogenic emissions ($27 \pm 15\%$). Fuzzy c-means clustering (FCM) was
applied to the afternoon time series of concentrations of $NO_y$, ozone, total particle number, black
carbon, and sulfate. Four clusters were identified and characterized by distinct airmass origins
and particle compositions. Two clusters, Bkgd-1 and Bkgd-2, were associated with background
conditions. Bkgd-1 appeared to represent near-field atmospheric PM production and oxidation of
a day or less. Bkgd-2 appeared to represent material transported and oxidized for two or more
days, often with out-of-basin contributions. Two other clusters, Pol-1 and Pol-2, represented the
Manaus influence, one apparently associated with the northern region of Manaus and the other



with the southern region of the city. A composite of the PMF and FCM analyses provided
insights into the anthropogenic effects on PM concentration and composition. The increase in
mass concentration of submicron PM ranged from 25% to 200% under polluted compared to
background conditions, including contributions from both primary and secondary PM.
Furthermore, a comparison of PMF factor loadings for different clusters suggested a shift in the
pathways of PM production under polluted conditions. Nitrogen oxides may have played a
critical role in these shifts. Increased concentrations of nitrogen oxides can shift pathways of PM
production from $HO_2$-dominant to NO-dominant as well as increase the concentrations of
oxidants in the atmosphere. Consequently, the oxidation of biogenic and anthropogenic precursor
gases as well as the oxidative processing of pre-existing atmospheric PM can be accelerated. The
combined set of results demonstrates the susceptibility of atmospheric chemistry, air quality, and
associated climate forcing to anthropogenic perturbations over tropical forests.



## 1. Introduction

Secondary organic material (SOM) constitutes a large fraction of the atmospheric particle burden (Hallquist et al., 2009; Jimenez et al., 2009) and therefore has important effects on the Earth's radiation balance, atmospheric visibility, and human health. SOM is a complex mixture of compounds resulting from many chemical pathways, and the processes underlying the production of SOM remain poorly understood. Models are especially challenged to accurately represent production of SOM in regions where there is a mix of biogenic and anthropogenic emissions (de Gouw et al., 2008; Glasius and Goldstein, 2016; Shrivastava et al., 2017). Possible shifts in the contributing mechanisms of SOM production between background and polluted conditions must be understood and quantified for distinct environments on the globe to test and enable accurate modeling predictions.

Several field observations, mainly in mid-latitudes of the Northern Hemisphere, and modeling efforts have suggested that the production of SOM from biogenic precursor compounds becomes more efficient in polluted air (Weber et al., 2007; Goldstein et al., 2009; Hoyle et al., 2011; Huang et al., 2014; Zhang et al., in press). In the northeastern USA, de Gouw et al. (2005) showed that organic PM concentrations correlated well with anthropogenic tracers, yet the concentrations of anthropogenic precursors were insufficient to explain the observed PM. In the southeastern USA, observations suggested that organic PM was produced mainly from BVOCs, however modulated by anthropogenic emissions of $NO_x$ and $SO_2$ (Weber et al., 2007; Goldstein et al., 2009). In the western USA, ground and aircraft measurements observed the highest organic PM increases when air masses having high concentrations of biogenic VOCs (BVOCs) intercepted anthropogenic emissions (Setyan et al., 2012; Shilling et al., 2013). A metastudy for several locations in the USA concluded that downwind urban air had increased



organic PM concentrations due to the photochemical production of SOM (De Gouw and
Jimenez, 2009). Models have estimated that 50 to 70% of the biogenic SOM mass concentration
in several locations is modulated by anthropogenic emissions (Carlton et al., 2010; Heald et al.,
2011; Spracklen et al., 2011). In addition, global-scale modeling studies have estimated an
increase of 20% to 60% in the global annual mean SOM concentration relative to the pre-
industrial period (Tsigaridis et al., 2006; Hoyle et al., 2009).

Many possible mechanisms may contribute to the effects of anthropogenic emissions on

increased SOM production, including changes in gas-particle partitioning, new particle
production and growth, and particle acidity. Changes in the concentrations of nitrogen oxides,
however, should be regarded as a critical factor (Hoyle et al., 2011 and references therein).
Different $NO_x$ regimes favor distinct gas-phase oxidation pathways, leading to different
oxidation products and particle yields, as evidenced in isoprene photo-oxidation (Kroll et al.,
2005, 2006; Hallquist et al., 2009; Worton et al., 2013; Liu et al., 2016b; Liu et al., 2016a). For
tropical forests, isoprene emissions are especially important in PM production (Martin et al.,
2010a; Chen et al., 2015). Under $HO_2$-dominant conditions (i.e., low $NO_x$), isoprene epoxydiols
(IEPOX) are produced in the gas phase and, through heterogenous reactions involving sulfate,
PM is produced (Paulot et al., 2009; Surratt et al., 2010). Depending on the relative importance
of increased concentrations of sulfate and $NO_x$ associated with pollution in a given region, an
enhancement or suppression of IEPOX-derived PM production relative to background conditions
may occur (Xu et al., 2015a; de Sá et al., 2017).

Amazonia, the largest tropical forest in the world and a large global source of SOM, is

comparatively understudied relative to northern mid-latitude regions, especially with respect to
the influence of pollution on the SOM lifecycle (Martin et al., 2010a). Manaus, a city of over two



million people in the central Amazon, continuously releases an urban plume into an otherwise
mostly unperturbed region (Kuhn et al., 2010; Martin et al., 2017). The region downwind of
Manaus, especially in the wet season in the absence of regional fires (Artaxo et al., 2013), offers
a natural laboratory for the investigation of biogenic-anthropogenic interactions and the resulting
consequences for the amount and composition of PM in the region. As part of GoAmazon2014/5,
de Sá et al. (2017) demonstrated that PM derived from IEPOX generally decreased under
polluted compared to background conditions downwind of Manaus. Nitrogen oxides in the
pollution plume suppressed the production of isoprene hydroxyhydroperoxides (Liu et al.,
2016b), leading to a decrease in the production of gas phase IEPOX and consequently of IEPOX-
derived PM (de Sá et al., 2017). IEPOX-derived PM was the exclusive focus of de Sá et al.

(2017).

The present study investigates the influences of urban pollution on the concentration and

composition of fine particles in central Amazonia, focusing on organic PM and its several
component classes. The analysis employs data sets collected during the first Intensive Operating
Period (IOP1) of the GoAmazon2014/5 experiment (Martin et al., 2016), corresponding to the
wet season during the period of February 1 to March 31, 2014. A separate publication is planned
for IOP2, corresponding to the dry season period of August 15 to October 15, when biomass
burning emissions were prevalent (de Sá et al., in preparation). Herein, positive-matrix
factorization (PMF) of organic mass spectra measured by aerosol mass spectrometry (AMS) in
conjunction with a clustering analysis of pollution indicators by Fuzzy c-means are employed to
investigate the changes in particle concentration and composition associated with the influence
of urban pollution downwind of Manaus.





## 2. Methodology

### 2.1 Site description

The primary site of this study, named "T3" (3.2133 °S, 60.5987 °W), was located 70 km

to the west of Manaus, Brazil, in central Amazonia (Martin et al., 2016). The site was situated in

a pasture of 2.5 km × 2 km surrounded by forest. Auxiliary sites "T0a" and "T0t", served as

references for background conditions in relation to T3 (Figure S1). Site T0a (2.1466 °S,

59.0050 °W) refers to the Amazonian Tall Tower Observatory (ATTO; Andreae et al., 2015),

located 150 km to the northeast of Manaus. Site T0t (2.5946°S, 60.2093°W) was situated 60 km

to the north-northwest of Manaus in the Cuieiras Biological Reserve ("ZF2") and refers to tower

"TT34", established in 2008 for the AMAZE-08 experiment (Martin et al., 2010b). The T0 sites

were typically upwind of Manaus, with only occasional transport of pollution to these sites

(Andreae et al., 2015; Chen et al., 2015). Auxiliary site "T2" served as a reference for polluted

conditions. This site was located just across the Rio Negro (3.1392°S, 60.1315°W), 8 km from

the southwestern edge of Manaus and typically downwind of urban emissions during the

daytime.

### 2.2 Aerosol Mass Spectrometry

Characterization of the atmospheric PM was obtained using a High-Resolution Time-of-

Flight Aerosol Mass Spectrometer (hereafter AMS; Aerodyne, Inc., Billerica, Massachusetts,

USA; DeCarlo et al., 2006; Canagaratna et al., 2007). Detailed aspects of the AMS operation in

GoAmazon2014/5 were presented in de Sá et al. (2017). In brief, the instrument was housed

within a temperature-controlled research container, and the inlet to the instrument sampled from

5 m above ground level. Ambient measurements for this study were obtained every other 4 min.



The other 4 min were used for analysis of output from an oxidation flow reactor as presented in
Palm et al. (2018).
Data analysis was performed using *SQUIRREL* (1.56D) and *PIKA* (1.14G) of the AMS
software suite (Sueper and collaborators; DeCarlo et al., 2006). Organic, sulfate, ammonium,
nitrate, and chloride PM mass concentrations were obtained from "V-mode" data. The choice of
ions to fit was aided by "W-mode" data, which were collected for one of every five days.
"Sulfate" and "nitrate" concentrations reported by the AMS may include contributions from both
organic and inorganic species (Farmer et al., 2010; Liao et al., 2015). Organic and inorganic
nitrate concentrations were estimated based on the ratio of $NO_2^+$ to $NO^+$ signal intensity, as
described in Section S2 (Fry et al., 2009; Farmer et al., 2010; Fry et al., 2013). The organic
elemental ratios, O:C and H:C, were calculated following the methods of Canagaratna et al.

(2015).

**2.3 Auxiliary measurements and datasets**
In complement to the AMS data set, the analysis herein incorporated auxiliary gas and
particle measurements from T3 (Martin et al., 2016). Mass concentrations of molecular and
tracer organic species in the gas and particle phases were measured by a Semi-Volatile Thermal
Desorption Aerosol Gas Chromatograph (SV-TAG) at a time resolution of one hour (Isaacman-
VanWertz et al., 2016). Concentrations of volatile organic compounds (VOCs) were measured
by a Proton-Transfer-Reaction Time-of-Flight Mass Spectrometer (PTR-ToF-MS; Liu et al.,
2016b). In the Mobile Aerosol Observing System (MAOS) of the ARM Climate Research
Facility (ACRF; Martin et al., 2016), measurements of $NO_y$ were made using a
chemiluminescence-based instrument (Air Quality Design). The raw $NO_y$ measurements (10-s
resolution) were averaged across 30-min intervals to dampen the influence of brief local events.



In addition, ozone concentrations were measured by an ultraviolet photometric analyzer (Thermo
Fisher, model 49i, Ozone Analyzer). Particle number concentrations were measured by a
Condensation Particle Counter (TSI, model 3772). Black carbon (BC) concentrations were
measured both by a 7-wavelength aethalometer (Magee Scientific, model AE-31) and a Single
Particle Soot Photometer (SP2; Droplet Measurement Techniques). The two datasets differed by
a factor of up to three in absolute mass concentrations, as observed in other studies (Subramanian
et al., 2007; Cappa et al., 2008; Lack et al., 2008), but they agreed well in the temporal trend.
The analysis herein for BC is thus restricted to the temporal trends. Wind direction, solar
irradiance, and precipitation rate were measured by the ARM Mobile Facility (AMF-1), which
was also part of the ACRF.

Additional measurements from T0a, T0t, and T2 were also used in the analysis. At T2,

non-refractory particle composition and concentration were measured by an Aerosol Chemical
Speciation Monitor (ACSM; Brito et al., in preparation). ACSM measurements were also made
at T0a during the wet season of 2015 (Carbone et al., in preparation). Further datasets collected
by AMS at T0t during the wet season of 2008 (AMAZE-08 campaign) were used in the analysis
(Chen et al., 2009;Schneider et al., 2011). AMS measurements made onboard the G-1 aircraft of
the ARM Aerial Facility (AAF) during IOP1 also supported the analysis herein (Shilling et al., in
preparation).
**2.4 Air mass backtrajectories and precipitation rates**

Simulations of two-day backward air mass trajectories, starting at 100 m above T3, were

made using HYSPLIT4 (Draxler and Hess, 1998). Input meteorological data were obtained
from the Global Data Assimilation System (GDAS), provided by the NOAA Air Resources


Laboratory (ARL), on a regular grid of 0.5° × 0.5°, 18 pressure levels, and 3-h intervals.
Trajectory steps were calculated for every 12 min.

Information on precipitation along the trajectories was obtained from the S-band radar of

the System for Amazon Protection (SIPAM) in Manaus (Machado et al., 2014). The radar had a
beam width of 1.8°, and it scanned 17 elevation angles every 12 min. Data were recorded to a
range of 240 km at 500-m gate spacing. The reflectivity fields were quality controlled to remove
non-meteorological echo and calibrated against the satellite precipitation radar of the Tropical
Rainfall Measuring Mission and Global Precipitation Measurement (TRMM-GPM; Kummerow
et al., 1998; Hou et al., 2014). Ground clutter was used to analyze the stability of the calibration.
The data were gridded at 2 km × 2 km in the horizontal and 0.5 km in the vertical using the
NCAR *Radx* software. The reflectivity at 2.5 km in altitude was converted to rain rates based on
the data sets of a Joss-Waldvogel disdrometer (Joss and Waldvogel, 1967), located at T3, 70 km
downwind of the radar.
**3. Results and discussion**
**3.1 Fine-mode PM composition**

The time series of mass concentrations of $PM_1$ species at T3 during the wet season of

2014 are plotted in Figure 1a. Organic material dominated the composition, contributing $79 \pm 7\%$
(average ± one standard deviation), followed by sulfate ($13 \pm 5\%$). The standard deviation
quantifies the variability across the time series. Average non-refractory (NR) $PM_1$ mass
concentrations and compositions at T3 as well as at three other sites in the region are represented
in Figure 1b. The two T0 sites corresponded to predominantly background conditions. By
contrast, the T2 site represented conditions just downwind of Manaus, and depending on wind
direction experienced fresh Manaus pollution or background air. The comparison in Figure 1b



demonstrates that the organic component consistently constituted 70% to 80% of NR-PM$_1$ across
sites in this region in the wet season, for both background and polluted conditions, in line with
previous observations (Chen et al., 2009; Martin et al., 2010a).

Even as the relative composition was similar across all sites, there were differences in the

absolute mass concentrations (Figure 1b, top panel). The NR-PM$_1$ mass concentrations at the T0
sites upwind of Manaus were approximately 1 μg m$^{-3}$. The concentrations at the T2 site just
downwind of Manaus were more than three times higher on average (3.3 μg m$^{-3}$). Average
concentrations at the T3 site (1.7 μg m$^{-3}$), several hours downwind of Manaus, were lower
compared to those at T2. This relative progression from T0, to T2, and then to T3 can be
understood as a first-order quantification of the overall effect of Manaus emissions in increasing
the airborne PM burden in the downwind region.

The diel trends of organic and sulfate mass concentrations across the four sites are shown

in Figure 2. Lines represent means, solid markers show medians, and boxes span interquartile
ranges. Organic mass concentrations and associated variability were higher at the T3 site
compared to the T0 sites, markedly so in the afternoon hours. The greater variability at T3 is in
line with a time-varying influence of Manaus emissions. This influence waxes and wanes with
small northerly or southerly shifts of the trade winds as well as other changes in regional
circulation tied to daily meteorology (Cirino et al., submitted). The higher afternoon mass
concentrations at T3 can be attributed to a combination of (i) an oxidant-rich, sunlight-fed plume
that increases the production rate of secondary PM and (ii) faster near-surface winds during the
day that transport PM from Manaus to T3 with less loss by deposition and dispersion compared
to more-stagnant air at night. Among all sites, the T2 observations had both the highest average
organic mass concentrations and the largest variability. These characteristics of the T2 dataset



can be explained by a combination of (i) the proximity of the site to Manaus, (ii) the rapid and
180° changes in wind direction caused by the intersection of the trade winds with a local river
breeze (dos Santos et al., 2014), and (iii) possible contributions of emissions from brick kilns,
located mostly southwest of the site, especially during night time (Martin et al., 2016; Cirino et
al., submitted).

The diel trends of the sulfate mass concentrations were in large part similar to those of

the organic mass concentrations. One distinction in the case of sulfate, however, is that the
variability at the T0 sites is similar to that at the T3 site. The explanation is that the background
sources of sulfate, including not only in-basin emissions but also out-of-basin long-range
transport, are variable and significant enough to make the variability at the background sites
similar to that at the T3 site (de Sá et al., 2017).

Overall, the organic $PM_1$ at T3 was highly oxidized, as indicated by the position of gray

markers in the plot of Figure 3. By contrast, the blue markers represent the dataset collected at
T2 during the same period. The datasets encompass all times of days and all conditions at both
sites. Datasets from background sites collected in different years are shown in Figure S2. Points
to the upper left represent more oxidized material, and points to the lower right represent less
oxidized material (Ng et al., 2011a). The comparison depicted in Figure 3 illustrates the effects
of the plume over the 4 h of transport from T2 to T3 (Cirino et al., submitted). The plot suggests
that the enhanced oxidative cycle associated with higher OH and $O_3$ concentrations in the
pollution plume might cause (i) the production of highly oxidized SOM, from both biogenic and
anthropogenic precursors including aromatic compounds (Chhabra et al., 2011; Lambe et al.,
2011), and (ii) the accelerated oxidative processing of pre-existing organic PM by OH and $O_3$





(Martin et al., 2017). The implication is that the emissions from Manaus can significantly affect
the mechanisms that produce or modify fine-mode PM over the tropical forest.
**3.2 Characterization of organic PM by positive-matrix factorization**
Positive-matrix factorization was applied to the time series of the organic component of
the high-resolution "V mode" mass spectra (Ulbrich et al., 2009b). Diagnostics of the PMF
analysis are presented in the Supplement (Section S1; Figures S3 and S4). Herein, "factor
profile" and "factor loading" refer to the mathematical products of the multivariate statistical
analysis, whereas "mass spectrum" and "mass concentration" refer to direct measurements.
A six-factor solution was obtained based both on the numerical diagnostics of the PMF
algorithm and the judged scientific meaningfulness of the resolved factors (Section S1). The
factor profiles, diel trends of the factor loadings, and the time series of the factor loadings and
other related measurements are plotted in Figures 4a, 4b, and 4c, respectively. The inset of
Figure 4a shows the mean fractional loading contribution of each factor during the analysis
period. The correlations of factor loadings with co-located measurements of gas- and particle-
phase species are shown in Figure 5.
The scientific interpretation of each factor was based on a combination of (i) the
characteristics of the factor profile (i.e., "mass spectrum"), as referenced to a worldwide database
of AMS spectra and PMF analyses (Ulbrich et al., 2009b; Ulbrich et al., 2009a, 2009c), and (ii)
the temporal correlations between the factor loading and other co-located measurements. Three
factors interpreted as primary emissions of organic PM were resolved: an anthropogenic-
dominated factor (hereafter, "ADOA"), a biomass burning factor ("BBOA"), and a fossil-fuel
hydrocarbon-like factor ("HOA"). Three factors interpreted as secondary production and



processing were resolved: a more-oxidized oxygenated factor ("MO-OOA"), a less-oxidized
oxygenated factor ("LO-OOA"), and an isoprene epoxydiols-derived factor ("IEPOX-SOA").

The HOA factor profile had characteristic ions of $C_4H_7^+$ and $C_4H_9^+$ at nominal values of

$m/z$ 55 and 57, respectively (Figure 4a). It had an oxygen-to-carbon (O:C) ratio of $0.18 \pm 0.02$,
the lowest among the six factors (Table 1). In line with the AMS PMF literature, the HOA factor
represents a class of primary hydrocarbon-like organic compounds that are typically associated
with traffic emissions (Zhang et al., 2005). In the present study, the HOA factor loadings
accounted for 6% of the organic mass concentrations on average (Figure 4a, inset). As a point of
comparison, the average in the southeastern USA typically varies from 9% to 15% (Xu et al.,
2015b). The lower relative contribution of 6% in this study might in part be due to a larger
relative role of secondary production in the environment of a tropical forest. In addition, the
distance from Manaus to the T3 site might allow time for substantial vertical mixing, dilution,
and subsequent evaporation of primary emissions into entrained background air (Robinson et al.,
2007; Liu et al., accepted; Shilling et al., in preparation). Finally, the possible differences in
emission profiles associated with different types of regional economic development between the
Brazilian Amazon and USA (e.g., fleet density, fuel matrix, industry, and so forth) should also be
considered. The HOA factor loading decreased during the day, which can be explained by the
growth of the planetary boundary layer (PBL) and the subsequent dilution of the concentrations
of primary emissions (Figure 4b). The time series of HOA factor loading did not correlate well
($R < 0.5$) with any of the co-located measurements at T3 (Figure 5). It is plotted alongside the
time series of $NO_y$ concentration in Figure 4c.

The BBOA factor profile was characterized by distinct peaks of $C_2H_4O_2^+$ ($m/z$ 60) and

$C_3H_5O_2^+$ ($m/z$ 73), as shown in Figure 4a. These peaks can be attributed to levoglucosan and



other anhydrous sugars that result from biomass pyrolysis (Schneider et al., 2006; Cubison et al.,
2011). Correlations of the factor loadings with the mass concentrations of levoglucosan and
vanillin ($R > 0.8$) measured by SV-TAG corroborate the association with biomass burning
(Figure 4c). The BBOA factor of this study had an O:C ratio of $0.61 \pm 0.08$ (Table 1), which is
consistent with large contributions from levoglucosan (O:C of 0.83) and similar sugars. The
factor loading on average accounted for 9% of the organic $PM_1$ mass concentration (Figure 4a,
inset). This result is consistent with the low incidence of fires in the Amazon during the wet
season (Martin et al., 2016). The BBOA factor loading typically decreased during the day
(Figure 4b), which is suggestive of the dilution of local sources during the development of the
PBL rather than long-range transport. Emissions from local fires around T3, including trash and
tree burning, as well as from wood-fueled brick kilns along the road from Manaus to T3 might
have contributed to this factor.

The ADOA factor profile, distinguished prominently by the $C_7H_7^+$ ion at $m/z$ 91, also had

characteristic ions of $C_4H_7^+$ at $m/z$ 55 and $C_3H_5^+$ at $m/z$ 41 (Figure 4a). A peak at $m/z$ 91 can arise
from many sources, including biogenic and anthropogenic emissions (Ng et al., 2011b). In itself,
$m/z$ 91 therefore does not serve as a tracer for a specific source or process without consideration
of the atmospheric context. Factors having a characteristic $m/z$ 91 peak (usually labeled "91fac")
typically have been associated with biogenic emissions (Robinson et al., 2011; Budisulistiorini et
al., 2015; Chen et al., 2015; Riva et al., 2016). The ADOA factor profile of this study, however,
more strongly resembles the mass spectra previously reported for PM emissions from cooking
activities (Lanz et al., 2007; Mohr et al., 2012) than those from "91fac" (Section S1; Figure S5).
The ratio of $m/z$ 55 to $m/z$ 57 of the ADOA factor was 4.1. This ratio lies in the range of 2 to 10
reported for several factors representing primary cooking emissions and is well above the range





of 0.8 to 1.4 reported for factors associated with traffic emissions, i.e., HOA (Mohr et al., 2012
and references therein; Hu et al., 2016). Even though the ADOA factor profile has a large
contribution from non-oxygenated ions, similar to HOA and consistent with a dominance by
primary emissions, it also contains considerable signal from oxygenated ions, resulting in a
relatively higher O:C of $0.40 \pm 0.05$ (Table 1). The factor loading on average accounted for 13%
of the organic $PM_1$ mass concentration (Figure 4a, inset). The factor loading decreased as the
PBL developed during the day, consistent with dominant non-photochemical, primary sources
(Figure 4b). Furthermore, there were increases, albeit small, in factor loading at 12:00 and 18:00
(local time), suggestive of breakfast-time and lunch-time cooking activities in Manaus based on a
transport time of 4 to 6 h between the city and the T3 site (Martin et al., 2016; Cirino et al.,
submitted). Manaus typically has four rush-hour periods each day from 6:30 to 8:00, 12:00 to
13:30, 16:30 to 18:30, and 21:00 to 22:00. Traffic peaking at these hours may therefore also have
contributed to the ADOA factor. Correlations between factor loading and external measurements
exceeded $R = 0.5$ for many anthropogenic markers, including concentrations of aromatics (e.g.,
benzene, toluene, and $C_8$ and $C_9$ species), carbon monoxide, particle count, and $NO_y$ (Figure 4c,
Figure 5). Contributions from secondary processes cannot be ruled out, and it is possible that PM
production from anthropogenic VOCs might have also been captured in this factor. Overall, the
ADOA factor was interpreted in the present study as an indicator of anthropogenic influence
associated with several sources in Manaus, most importantly cooking and possibly traffic
emissions.

The IEPOX-SOA factor profile had marker ions of $C_4H_5^+$ (*m/z* 53) and $C_5H_6O^+$ (*m/z* 82)

(Figure 4a; Lin et al., 2012; Hu et al., 2015; de Sá et al., 2017). It had an O:C ratio of $0.9 \pm 0.10$
(Table 1). The factor loading on average accounted for 17% of the organic $PM_1$ mass



concentration (Figure 4a, inset). There were high correlations ($R > 0.8$) between factor loadings
and concentrations of $C_5$-alkenetriols and 2-methyltetrols, which are markers of IEPOX-derived
PM, produced by the photo-oxidation of isoprene under $HO_2$-dominant conditions (Surratt et al.,
2010; Lin et al., 2012; Figure 4c). The increase in factor loading during daytime was consistent
with a photochemical source (Figure 4b). There were also correlations between factor loadings
and concentrations of sulfate and some acids, such as tricarballylic acid (TCA; Figure 5), in
agreement with the association of IEPOX-derived PM and sulfate/acidity observed in other
studies (Budisulistiorini et al., 2013; Nguyen et al., 2014; Kuwata et al., 2015). Overall, this
factor was therefore interpreted as representative of PM produced from isoprene photo-oxidation
under $HO_2$-dominant conditions. The effects of urban pollution on the loadings of this factor
were the focus of a previous publication (de Sá et al., 2017).

The two remaining factors, LO-OOA and MO-OOA, were also associated with secondary

atmospheric processes. The LO-OOA and MO-OOA factors had O:C ratios of $0.72 \pm 0.10$ and
$1.09 \pm 0.17$, respectively. The LO-OOA factor was characterized by the greatest ratio of signal
intensity of the $C_2H_3O^+$ ion ($m/z$ 43) to that of the $CO_2^+$ ion ($m/z$ 44) (Figure 4a) compared to all
other factors. This factor is usually attributed to lower-generation, less-oxidized, higher-volatility
secondary organic PM (Jimenez et al., 2009). By comparison, the MO-OOA factor profile had
the strongest $CO_2^+$ ($m/z$ 44) peak among all factors (Figure 4a). This factor is usually attributed
to higher-generation, more-oxidized, less-volatile secondary organic PM or extensively oxidized
primary PM of any type that has resided in the atmosphere for several days or more (Jimenez et
al., 2009).

The LO-OOA factor loading on average accounted for 25% of the organic $PM_1$ mass

concentration (Figure 4a, inset). The factor loading correlated better with the estimated





concentrations of inorganic nitrate than with organic or total nitrate (Figure 5; Section S2 and
Figure S6), which is consistent with the interpretation of the higher volatility associated with this
factor (Jimenez et al., 2009; Zhang et al., 2011). The factor loading also correlated ($R > 0.7$) with
the concentrations of 2-methylglyceric acid and methyl-butyl-tricarboxylic acid (MBTCA),
which are products of isoprene and monoterpene oxidation, respectively, under NO-dominant
conditions (Figure 4c; Figure 5). The factor loading increased starting at 9:00 (local time) and
peaked in the afternoon hours (Figure 4b). This diel trend, tied to the sunlight cycle, tracked the
typical daily emission patterns of isoprene and monoterpenes from the surrounding forest
(Yáñez-Serrano et al., 2015). The absence of a sharp decline at sunset and the higher variability
at nighttime may also indicate a contribution by terpene ozonolysis. For these several reasons,
the LO-OOA factor was interpreted herein as secondary organic PM produced mostly within
several hours of observations by many possible pathways, including (i) the photo-oxidation of
isoprene along non-IEPOX pathways, (ii) the photo-oxidation of terpenes and other biogenic
VOCs along both $HO_2$- and NO-dominant reaction pathways, (iii) the ozonolysis of terpenes, and
(iv) the possible production of SOM from anthropogenic emissions from Manaus.

The MO-OOA factor loading on average accounted for 30% of the organic $PM_1$ mass

concentration (Figure 4a, inset). The factor loading correlated ($R > 0.7$) with the mass
concentrations of several particle-phase carboxylic acids as well as the concentrations of sulfate,
ammonium, and ozone (Figure 5). The time series of malic acid and ozone concentrations are
shown alongside the MO-OOA factor loadings in Figure 4c. Malic acid is a highly oxidized
compound (O:C of 1.25), which may have many different sources (Röhrl and Lammel, 2002; van
Pinxteren et al., 2014). The MO-OOA factor loading increased starting at 8:00 (local time;
sunrise was at 6:00) and peaked between 10:00 and 16:00, with a large variability in the factor



loadings in the afternoon hours among different days (Figure 4b). The afternoon increase and
day-to-day variability were consistent with strong but variable photochemical processing leading
to further oxidation of organic PM during the day, depending on daily weather. The high O:C of
$1.09 \pm 0.17$ could also be indicative of production of PM from aromatic compounds emitted from
Manaus (Chhabra et al., 2011; Lambe et al., 2011). Overall, this factor was interpreted to
represent highly oxidized PM from multiple processes. Species initially associated with HOA,
BBOA, ADOA, IEPOX-SOA, and LO-OOA factors may converge after sufficient atmospheric
oxidation to become represented by the MO-OOA factor (Jimenez et al., 2009; Palm et al.,

2018).

**3.3 Shifts in PM with anthropogenic influences**
**3.3.1  Cluster Analysis**

To further investigate changes in the concentration and composition of PM associated

with anthropogenic influences, a Fuzzy c-means (FCM) algorithm was applied to the time series
of concentrations of particle number, $NO_y$, ozone, black carbon, and sulfate measurements at the
T3 site (Bezdek et al., 1984). The analysis was fully independent of the PMF results. For each
point in time, these concentrations represented the spatial coordinates of the data point. As
discussed below, four clusters were identified. Based on measures of spatial similarity, the
clustering algorithm attributed to each data point a degree of membership relative to each of the
four clusters (Section S3; Figure S7 and Figure S8).

The scope of the clustering analysis was restricted to afternoon time points for which ten-

hour airmass backtrajectories did not intersect significant precipitation and for which solar
irradiance at T3 averaged over the previous 4 h was higher than 200 W m$^{-2}$ (Section S3). This
scope aimed at capturing fair-weather conditions and thereby minimizing the role of otherwise



confounding processes, such as boundary layer dynamics and wet deposition. The elimination of
trajectories having precipitation, however, should not be regarded as fully accurate given the
uncertainties in the HYSPLIT trajectories. The scoped dataset spanned 24 afternoons.

Four clusters were identified based on minimization of the FCM objective function as

well as a subjective assessment of meaningful interpretation of the set of clusters (Section S3).
The FCM algorithm returned a matrix containing the degrees of membership (ranging from 0 to
1) to each of the four clusters (columns) for each point in time (rows). For any given time point
(i.e., row), the sum of its degrees of membership to clusters (i.e., sum across columns) was
always unity, by definition. A collection of examples, representing 37% of the analyzed data
points by FCM, is shown in Figure 6a. For times predominantly associated with only one cluster
(e.g., Feb 9 and Feb 10), the corresponding air mass backtrajectories are plotted in Figure 7. The
FCM algorithm also returned the coordinates of cluster centroids, which are listed in Table 2.

Two clusters of data were interpreted as "background" and labeled "Bkgd-1" and "Bkgd-

2". They were characterized by $NO_y < 1$ ppb, ozone < 20 ppb, and particle number < 1200 $cm^{-3}$
(Table 2; Figure 6). The two clusters differed especially in that Bkgd-2 had significantly larger
concentrations of sulfate and black carbon. A comparison of the datasets of Feb 13
(predominantly Bkgd-1) and Feb 16 (predominantly Bkgd-2) in Figure 6 highlights these
differences. Concentrations of sulfate and black carbon were 0.15 and 0.10 $\mu g\ m^{-3}$, respectively,
on Feb 13, compared to 0.40 and 0.15 $\mu g\ m^{-3}$ on Feb 16. The backtrajectories associated with
Bkgd-1 had both northeasterly and southeasterly components. The wind fields, out of line with
the trade winds, may suggest passage through recent weather systems and may imply wet
deposition, which in turn might explain lower gas and particle concentrations (Table 2). These
recent weather systems might not have been excluded from the scoped dataset because of



inaccuracies in the intersections of the backtrajectories with precipitation data, as discussed
above, or because they were more distant than captured by the 10-h backtrajectories. Consistent
with this hypothesis, the centroid value calculated for the 4-h averaged solar irradiance at T3
(Section 3.3.2) was lower for Bkgd-1 (400 W m$^{-2}$) compared to the other clusters (600 W m$^{-2}$),
suggesting an association of Bkgd-1 with overcast conditions. By comparison, the
backtrajectories associated with Bkgd-2 were predominantly from the northeast, coming from the
direction of the T0t and T0a sites (Figure 7), in line with the dominant trade winds of the wet
season. The air masses of Bkgd-2 may have experienced less wet deposition and may represent
more extensive atmospheric oxidation than those of Bkgd-1. They may also have carried PM
contributions from out-of-basin sources, which would be consistent with the higher sulfate and
black carbon concentrations of Bkgd-2 compared to Bkgd-1 (Chen et al., 2009; Pöhlker et al.,

2017).

Two other clusters were interpreted as "polluted" and labeled "Pol-1" and "Pol-2". They

were characterized by concentrations of NO$_y$ > 1 ppb, ozone > 20 ppb, and particle number >
1200 cm$^{-3}$ (Table 2; Figure 6). The dataset of the afternoon of Mar 9 illustrates a shift in
dominance from Pol-2 to Pol-1 (Figure 6). Although Pol-1 and Pol-2 both have high
concentrations of sulfate and other pollutants, they differ in the extent of those high
concentrations. The explanation may be that these clusters represent different source regions.
Pol-1 may be associated with emissions from the northern region of Manaus, and Pol-2 may be
associated with emissions from the southern region of Manaus. Industry, power production, and
oil refineries are concentrated in the southeastern region of Manaus (Figure S9; Medeiros et al.,
2017). Population density and commercial activity is concentrated in the southwestern portion of
the city where downtown is located (Figure S10).  Aircraft observations show that concentrations





of sulfate as well as other pollutants are higher in the urban outflow from the southern compared
to the northern region of Manaus (Figure S10). Directional plots of $SO_2$ and particle number
concentrations observed at the T2 site further demonstrate the heterogeneity in Manaus
emissions (Figure S10). This hypothesis of a geographical difference in source regions
qualitatively aligns with the differences in backtrajectories characteristic of times dominated by
Pol-1 and Pol-2 (Figure 7). This interpretation does imply, however, that the backtrajectories
may have a 20° inaccuracy. Such inaccuracy is reasonable for the application of HYSPLIT
modeling in this region given (i) the absence of surface weather stations and (ii) the relatively
large scale of input wind fields (i.e., 50 km) compared to the scale of modeling (i.e., 70 km from
T3 to Manaus and a city cross section of 20 km).
**3.3.2 Comparison of PM among clusters**

The characteristic PM composition associated with each cluster was determined by

calculating the centroid coordinates of the clusters for the AMS species and PMF factors
(Section S3). The centroid coordinate of a cluster for a given variable is defined as a weighted
mean of that variable across all points in time, where the weight is the degree of membership of
each data point to that cluster. A comparison of $PM_1$ concentrations and compositions for the
four clusters is shown in Figure 8. Values are listed in Table 2.

The NR-$PM_1$ mass concentrations increased by 25% to 200% in clusters Pol-1 and Pol-2

compared to clusters Bkgd-1 and Bkgd-2 (Figure 8a). Increases in sulfate and associated
ammonium concentrations had a smaller yet non-negligible role in the increased $PM_1$ mass
concentrations. Sources of sulfate other than Manaus sustain relatively high concentrations in the
Amazon basin, as represented by the Bkgd-2 cluster (Chen et al., 2009; de Sá et al., 2017).
Compared to these regional background concentrations (i.e., Bkgd-2 cluster), the increases in





sulfate concentrations were significant only for air masses associated with the heavily
industrialized and densely populated southern region of Manaus (i.e., Pol-2 cluster).

With respect to the composition of the organic PM, Figure 8b shows that the Bkgd-1

cluster had large contribution from the LO-OOA factor. By comparison, the Bkgd-2 cluster had
larger contributions from the MO-OOA and IEPOX-SOA factors. A comparison of 13 Feb and
16 Feb of 2014 (Figure 6d) illustrates these findings. The low mass concentrations and the
dominant contribution by the LO-OOA factor suggest that the Bkgd-1 cluster may represent
conditions under which secondary organic PM was produced within recent hours through photo-
oxidation of VOCs emitted by the forest and subsequent condensation of secondary organic
material. The low sulfate concentrations for Bkgd-1 may rationalize the absence of a significant
contribution by the IEPOX-SOA factor. Isoprene photo-oxidation may have contributed to PM
production by pathways other than IEPOX uptake (Krechmer et al., 2015; Riva et al., 2016). By
comparison, for Bkgd-2, the higher mass concentrations and the greater contributions by IEPOX-
SOA and MO-OOA factors suggest that this cluster may represent conditions under which
secondary organic PM was a combination of material produced both on that day as well as on
previous days. During transport, the organic PM may have undergone extensive atmospheric
oxidation by a combination of surface and condensed-phase chemistry, including cloud water
processes (Carlton et al., 2006; Ervens et al., 2011; Hoyle et al., 2011; Perraud et al., 2012).
Concentrations and composition of the Bkgd-2 cluster may therefore represent an extensive
geographical footprint.

The organic PM concentration and composition associated with the Pol-1 and Pol-2

clusters were distinct from those of the Bkgd-1 and Bkgd-2 clusters (Figure 8). The mass
concentrations of organic PM were greater by 25% to 150% for Pol-1 and Pol-2. According to





the PMF factors (Figure 8b), the larger part of this increase in organic PM between the
background and polluted clusters was tied to the production of secondary organic PM, although
primary emissions also contributed significantly. By comparison, for both Bkgd-1 and Bkgd-2
clusters, contributions by primary emissions were negligible, as indicated by the low summed
contribution of factors of primary origin (i.e., ADOA, BBOA, and HOA) to the organic $PM_1$ (<
10%). For Pol-1 and Pol-2, the ADOA factor loading on average accounted for 10% of the
organic mass concentration at T3, serving as a strong marker of Manaus pollution. A comparison
of 9 Feb and 9 Mar with 13 Feb and 16 Feb illustrates these findings (Figure 6d).

In regard to secondary organic PM, the IEPOX-SOA factor loading decreased by almost

50% under polluted compared to background conditions. de Sá et al. (2017) attributed this
decrease to the suppression of IEPOX production by elevated NO concentrations. This
suppression typically outweighed possible enhancements in IEPOX uptake and subsequent PM
production because of elevated sulfate concentrations. By contrast, the LO-OOA and MO-OOA
factor loadings increased by 50% to 100% under polluted conditions. These increases exceeded
the decrease in IEPOX-SOA factor loadings, resulting in a net increase of around 100% in mass
concentration of secondary organic PM (Figure 8).

The shifts in the processes governing the production of secondary organic PM because of

increased $NO_x$, OH, and $O_3$ concentrations characteristic of the pollution plume were complex
and non-linear (Figure 9a). Overall, the oxidation pathways were driven faster. The relatively
high $f_{CO_2^+}$ values and O:C ratios of all factors (Table 1), including those associated with primary
emissions, compared to typical values at other locations worldwide (Canagaratna et al., 2015),
corroborate this interpretation. Ozone concentrations in the plume increase by 200 to 300 %, and
hydroxyl radical concentrations increased by 250% or more (Liu et al., accepted). As $HO_2$-



dominant pathways were inhibited, NO-dominant pathways became active. Increased oxidant
concentrations may also have promoted additional multigenerational chemistry of semi- or
intermediate-volatility species (Robinson et al., 2007). Oxidation of VOCs by aqueous-phase
reactions, including in-cloud processing, and oxidation of biomass burning emissions may also
have played roles to varying degrees on different days (Carlton et al., 2006; Ervens et al., 2011;
Hoyle et al., 2011; Perraud et al., 2012). In addition, when primary and secondary PM mass
concentrations increased, further uptake of oxidized semi-volatile molecules could have been
thermodynamically favored according to partitioning theory, representing a positive feedback on
the increase of mass concentrations (Pankow, 1994; Odum et al., 1996; Carlton et al., 2010).
The increase in the LO-OOA and MO-OOA factor loadings associated with Pol-1 and
Pol-2 indicates that the net effect of this accelerated and modified chemistry was the quick
production and further oxidation of secondary organic PM. Precursors may have included both
the wide range of biogenic VOCs as well as contributions from anthropogenic precursors, such
as gas-phase species from vehicle emissions or evaporated primary material (Nordin et al., 2013;
Presto et al., 2014). The LO-OOA factor loading was important for the polluted conditions of
Pol-1 and Pol-2 as well as for the clean conditions of Bkgd-1. This result is not necessarily
because of an in-common molecular composition but rather because of an in-common process,
i.e., fresh production of secondary organic PM (Figure 9b). Likewise, the MO-OOA factor
loading was important for Pol-1, Pol-2, and Bkgd-2 because this factor represented an in-
common process, i.e., extensive oxidation (Figure 9b). In the case of the MO-OOA factor, there
is also an overall in-common composition characterized by highly oxidized species even as
precursor species and subsequent oxidation pathways differed (Jimenez et al., 2009).



The complexity of the real atmospheric processes, as illustrated in Figure 9, is to some

extent captured by the instrumental and analytical tools herein employed. Positive-matrix

factorization identified several broad classes of organic PM. Some PMF factors had sufficiently

unique signatures that they could be associated to one specific source and/or process (e.g., HOA

and IEPOX-SOA). Other factors, in contrast, represented a wide range of sources that shared in-

common processes (e.g., LO-OOA and MO-OOA). The clustering analysis contextualized the

PMF results and demonstrated that the effects of the urban pollution were neither limited to nor

captured by a single PMF factor. Instead, the urban plume influenced several PMF factors in

different ways and to different extents. The implication is that changes in the AMS spectral

signature of the organic PM caused by polluted conditions may not be sufficiently unique to

allow for its complete separation by PMF analysis alone, especially in respect to the production

of secondary organic PM. In this context, the Fuzzy c-means analysis served herein as a useful

tool to incorporate auxiliary datasets and thereby to further understand anthropogenic influences

on PM production and characteristics.

**4. Summary and conclusions**

Changes in the concentrations and the composition of fine-mode PM due to the influence

of anthropogenic emissions were investigated for the Amazonian wet season. Organic material

dominated the submicron composition, consistently representing between 70% and 80% of the

$PM_1$ mean mass concentration across measurement sites upwind and downwind of Manaus and

across different levels of pollution. Absolute mass concentrations, however, varied significantly

among sites. Average concentrations downwind of Manaus were 100% to 200% higher than

those upwind. Furthest downwind at T3, the organic component was more oxidized compared to

that at the T2 site.



Positive-matrix factorization and Fuzzy c-means clustering were applied to the datasets to
obtain a composite analysis of the shifts in $PM_1$ concentrations and composition under polluted
conditions. Based on the FCM clustering, every point in time at T3 was interpreted as being
affected by a combination of four influences, as represented by four clusters. Two background
(Bkgd-1 and Bkgd-2) and two polluted (Pol-1 and Pol-2) clusters were identified. Particle mass
concentrations were double for polluted compared to background conditions. Contributions from
secondary processes dominated (> 80%) for both background and polluted conditions.
In terms of primary emissions, absolute contributions increased by a factor of five or
more under polluted conditions, corresponding to an increase from < 10% to 15% of total $PM_1$.
The ADOA factor loading increased over five-fold for the polluted compared to the background
clusters, and this factor thus served as a strong tracer of Manaus pollution. BBOA and HOA
factor loadings, associated with biomass burning and fossil fuels, respectively, increased by two-
fold with pollution. The ADOA factor loading represented 61% to 76% of the total primary
factor loadings for the Pol-1 and Pol-2 clusters.
As for the secondary processes, the analysis further finds that the pollution plume acted
both to shift pathways of secondary organic PM production and to accelerate the atmospheric
oxidation of pre-existing organic PM. The oxidation of biogenic PM precursors shifted from
$HO_2$- to NO-dominant pathways, and the oxidation of anthropogenic precursors possibly
contributed to increased PM concentrations. The IEPOX-SOA factor loadings were highest for
the Bkgd-2 cluster, associated with long-range transport under background conditions, and
decreased by almost 50% for the polluted clusters, in line with a shift of isoprene oxidation from
$HO_2$- to NO-dominant pathways. Concomitantly, the LO-OOA factor loading increased by more
than 50% for these clusters, suggesting rapid in-plume production of secondary organic PM





through several pathways. The LO-OOA factor was also important for the Bkgd-1 cluster,
associated with fresh background conditions, which is suggestive of recent biogenic organic PM
production. The MO-OOA factor had large relative contributions in the Bkgd-2, Pol-1, and Pol-2
clusters, suggestive of significant oxidative processing associated with these clusters. Increases
of up to 300% in the MO-OOA factor loadings for Pol-1 and Pol-2 relative to background
conditions of Bkgd-1 showed the effects of an accelerated oxidation cycle, leading to highly
oxidized PM downwind of Manaus. Based on this and related studies (Liu et al., 2016b; de Sá et
al., 2017; Martin et al., 2017), the critical lever seems to be increased concentrations of nitrogen
oxides in the pollution plume for both directly shifting and indirectly accelerating mechanisms of
secondary organic PM production in central Amazonia during the wet season.

The altered composition under anthropogenic influences also affects the physical

properties of the $PM_1$. Bateman et al. (2017), using the results of the PMF analysis presented
herein, reported a shift from predominantly liquid PM under background conditions to a
considerable presence of non-liquid PM above 50% RH under polluted conditions. Non-liquid
PM can have different reactive chemistry from liquid PM (Li et al., 2015; Liu et al., 2018). A
linear relationship between the increase in particle rebound fraction and the sum of ADOA,
BBOA, and HOA factor loadings had an $R^2$ of 0.7. The highest individual correlation was with
the ADOA factor loading (Bateman, personal communication). In addition, Thalman et al.
(2017), also using the PMF results reported herein, concluded that the larger relative contribution
of secondary organic material during the daytime compared to the nighttime was the primary
driver of the diel trend of higher particle hygroscopicity during the day compared to the night, as
tied to cloud condensation nuclei (CCN) properties.



This study communicates a snapshot of the changes that occur in the atmospheric
composition over a tropical forest because of regional urbanization. In the context of a forest in
transition (Davidson et al., 2012), the findings herein provide a quantitative assessment of the
effects of urban pollution on the forested surroundings of Manaus. The studied region and the
observed changes in atmospheric composition represent a microcosm that might become more
widespread through Amazonia as urbanization trends continue in the future. Further
investigations of the specific chemical pathways and physical mechanisms that enhance PM
production in the urban plume are warranted to understand what other pollutants are critical for
control in the context of ongoing and future air quality regulation in the study region as well as
for other tropical forested environments worldwide.



**Acknowledgments.** Institutional support was provided by the Central Office of the Large Scale

Biosphere Atmosphere Experiment in Amazonia (LBA), the National Institute of Amazonian

Research (INPA), and Amazonas State University (UEA). We acknowledge support from the

Atmospheric Radiation Measurement (ARM) Climate Research Facility, a user facility of the

United States Department of Energy (DOE, DE-SC0006680), Office of Science, sponsored by

the Office of Biological and Environmental Research, and support from the Atmospheric System

Research (ASR, DE-SC0011115, DE-SC0011105) program of that office. Additional funding

was provided by the Amazonas State Research Foundation (FAPEAM 062.00568/2014 and

134/2016), the São Paulo State Research Foundation (FAPESP 2013/05014-0), the USA

National Science Foundation (1106400 and 1332998), and the Brazilian Scientific Mobility

Program (CsF/CAPES). S. S. de Sá acknowledges support by the Faculty for the Future

Fellowship of the Schlumberger Foundation. BBP is grateful for a US EPA STAR Graduate

Fellowship (FP-91761701-0). The authors thank Paulo Castillo for his assistance in quality-

checking the black carbon data from MAOS. Data access from the Sistema de Proteção da

Amazônia (SIPAM) is gratefully acknowledged. The research was conducted under scientific

license 001030/2012-4 of the Brazilian National Council for Scientific and Technological

Development (CNPq).





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



**List of Figures**





The ordinate scale for the T2 panel differs from the other three panels. Concentrations were adjusted to standard temperature (273.15 K) and pressure ($10^5$ Pa).

**Figure 3**. Scatter plot of the AMS signal fraction at $m/z$ 44 ($f_{44}$) against that at $m/z$ 43 ($f_{43}$). Gray and blue circles correspond, respectively, to measurements at T3 and T2 during IOP1, in the wet season of 2014. Solid squares represent median values, and whiskers represent 10 and 90 percentiles. Dashed lines delineate the region where worldwide measurements of ambient organic $PM_1$ commonly lie (Ng et al., 2011a).

**Figure 4**. Results of the PMF analysis on the time series of AMS organic mass spectra collected at T3. (a) Mass spectral profile of each factor represented at unit mass resolution. The inset shows the mean fractional loading of each factor. (b) Diel trends for the loadings of each PMF factor. Local time is (UTC - 4 h). Lines represent means, solid markers show medians, and boxes span interquartile ranges. (c) Time series of the factor loadings (left axis) and other related measurements at T3 (right axis). Methyl-butyl-tricarboxylic acid is abbreviated as MBTCA.

**Figure 5**. Column plot of Pearson $R$ correlations between the loading of each PMF factor and values of selected measurements at T3. Abbreviations include tricarballylic acid (TCA), methyl-butyl-tricarboxylic acid (MBTCA), methyl vinyl ketone (MVK), methacrolein (MACR), and isoprene hydroxyhydroperoxides (ISOPOOH). SV-TAG measurements refer to particle-phase concentrations. Isomers could not be distinguished by PTR-ToF-MS measurements; $C_8$ and $C_9$ aromatics include the xylene and trimethylbenzene isomers, respectively.

**Figure 6**. Results of the cluster analysis by Fuzzy c-means (FCM) for afternoon periods (12:00 to 16:00 h) are presented by several case studies. (a) Degree of membership in each of



the four clusters. The sum of degrees of membership across all clusters is unity.
Background conditions are abbreviated as "Bkgd", and polluted conditions are
abbreviated as "Pol". (b) Pollution indicators: concentrations of $NO_y$, $O_3$, black carbon
(BC), and particle number count are plotted. (c) $PM_1$ mass concentrations for organic,
sulfate, nitrate, and ammonium species. (d) Fractional contribution of each factor to
total organic $PM_1$.

**Figure 7**. Air mass backtrajectories associated with the four clusters of the FCM analysis for the
case studies of Figure 6. Trajectories were calculated using HYSPLIT 4 in steps of 12
min for ten hours (Draxler and Hess, 1998). Image data: Google earth.

**Figure 8**. Characteristic PM composition of the FCM clusters as represented by coordinates of
cluster centroids. (a) Mass concentrations of AMS species characteristic of each
cluster. (b) PMF factor loadings characteristic of each cluster. Calculations are
presented in more detail in the Supplementary Material (Section S3). Values plotted
are shown in Table 2.

**Figure 9**. Schematic representation of (a) atmospheric processes, illustrated in a simplified
manner, associated with the production of organic $PM_1$ and (b) observables of these
processes as captured by the datasets and analytical approach employed in this study.
In panel (a), the left side depicts the emissions of biogenic volatile organic compounds
(VOCs), their atmospheric oxidation, and the production of biogenic secondary
organic $PM_1$. The right side depicts anthropogenic emissions of gas species and
particulate matter that can alter natural atmospheric concentrations and processes.
There are primary organic $PM_1$ emissions from traffic, cooking, and industrial





activities. Anthropogenic VOCs can be precursors for the production of secondary

organic $PM_1$ and can affect the production of ozone and hydroxyl radical. $NO_x$

emissions directly and indirectly alter the natural pathways of $PM_1$ production in the

atmosphere. $NO_x$ and $SO_x$ can also directly contribute to the formation of secondary

inorganic $PM_1$ (not shown), which can in turn play a role in changing pathways of

secondary organic $PM_1$ production. In panel (b), different PMF factors represent

distinct sources and/or processes. The IEPOX-SOA factor is at the intersection of the

two, as it represents both a source (i.e., isoprene emissions from the forest) and a

process (i.e., photo-oxidation under $HO_2$ dominant conditions, influenced by sulfate

concentrations). The dashed black line represents the natural and anthropogenic

oxidative processes that transform the chemical signature of the HOA, ADOA, BBOA,

IEPOX-SOA, and LO-OOA factors after sufficient atmospheric residence time into the

MO-OOA factor. The clusters represent different conditions at the receptor site (i.e.,

T3) and therefore incorporate the meteorological and geographical histories of the air

masses that reach the site and affect the observed concentrations. The different PMF

factors are associated to the different clusters (solid lines) to various extents (not

detailed here for simplification purposes; cf. Figure 8).



**Table 1.** Characteristics of the PMF factors derived from the AMS datasets. Listed are signal fractions $f_{CO_2^+}$ at nominal $m/z$ 44 and oxygen-to-carbon (O:C) and hydrogen-to-carbon (H:C) ratios. Values and associated uncertainties were calculated by running PMF in "bootstrap mode" (Ulbrich et al., 2009b). Elemental ratios were calibrated by the "improved-ambient" method, which has an estimated uncertainty of 12% for O:C and 4% for H:C (Canagaratna et al., 2015).

| PMF factor | $f_{CO_2^+}$ | O:C | H:C |
|---|---|---|---|
| MO-OOA | 0.25 ± 0.01 | 1.09 ± 0.17 | 1.27 ± 0.12 |
| LO-OOA | 0.14 ± 0.02 | 0.72 ± 0.10 | 1.49 ± 0.07 |
| IEPOX-SOA | 0.17 ± 0.01 | 0.93 ± 0.10 | 1.39 ± 0.07 |
| ADOA | 0.11 ± 0.01 | 0.40 ± 0.05 | 1.63 ± 0.02 |
| BBOA | 0.123 ± 0.004 | 0.61 ± 0.08 | 1.57 ± 0.04 |
| HOA | 0.048 ± 0.006 | 0.18 ± 0.02 | 1.94 ± 0.02 |



**Table 2.** Coordinates of cluster centroids for input variables, AMS species concentrations, and PMF factor loadings. Table entries for AMS species and PMF factors are plotted in Figure 8. The AMS species concentrations (except for sulfate) and PMF factor loadings were not used as input variables in the FCM clustering analysis.

| Species | Cluster Centroid | | | |
|---|---|---|---|---|
| | Bkgd-1 | Bkgd-2 | Pol-1 | Pol-2 |
| **Input variables** | | | | |
| Particle number ($cm^{-3}$) | 714 | 1117 | 2636 | 6697 |
| $NO_y$ (ppb) | 0.64 | 0.95 | 1.2 | 2.2 |
| $O_3$ (ppb) | 14 | 17 | 26 | 36 |
| Black carbon ($\mu g\ m^{-3}$) | 0.05 | 0.16 | 0.21 | 0.18 |
| Sulfate ($\mu g\ m^{-3}$) | 0.15 | 0.36 | 0.44 | 0.57 |
| **AMS species concentrations ($\mu g\ m^{-3}$)** | | | | |
| Organic | 0.96 | 2.0 | 2.5 | 2.6 |
| Ammonium | 0.05 | 0.12 | 0.15 | 0.21 |
| Nitrate | 0.03 | 0.07 | 0.10 | 0.12 |
| Chloride | 0.007 | 0.011 | 0.009 | 0.007 |
| **PMF factor loadings ($\mu g\ m^{-3}$)** | | | | |
| MO-OOA | 0.29 | 0.83 | 1.13 | 1.13 |
| LO-OOA | 0.38 | 0.41 | 0.62 | 0.77 |
| IEPOX-SOA | 0.18 | 0.49 | 0.43 | 0.29 |
| ADOA | 0.044 | 0.086 | 0.19 | 0.32 |
| BBOA | 0.028 | 0.054 | 0.081 | 0.063 |
| HOA | 0.017 | 0.027 | 0.039 | 0.040 |





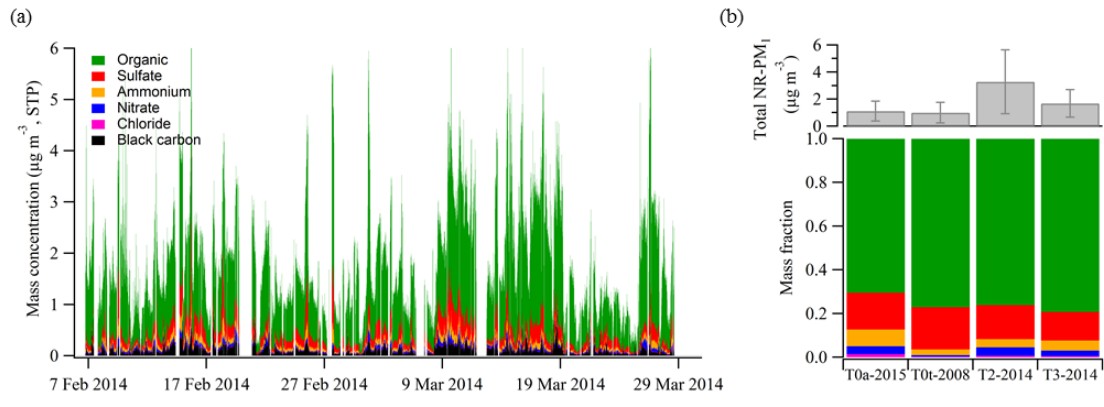

*Figure 1*

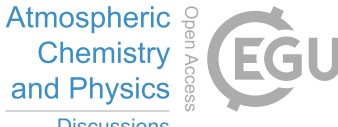

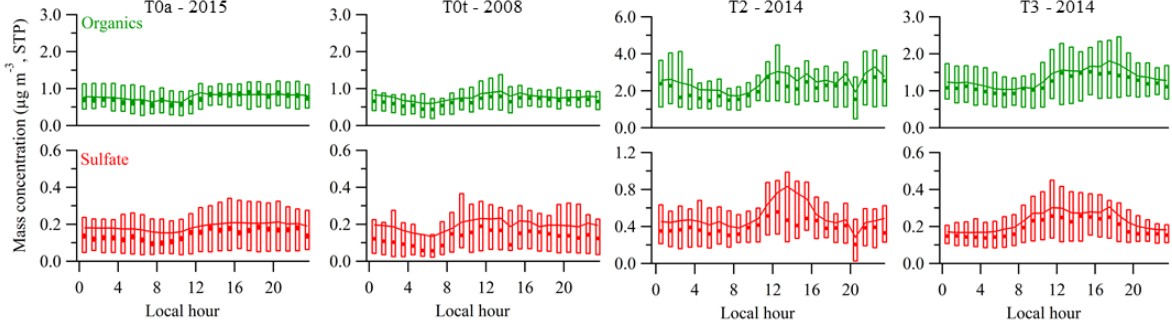

*Figure 2*



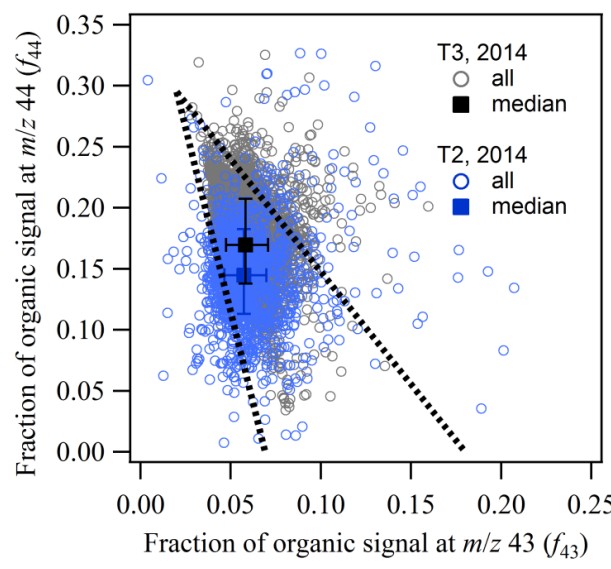

*Figure 3*





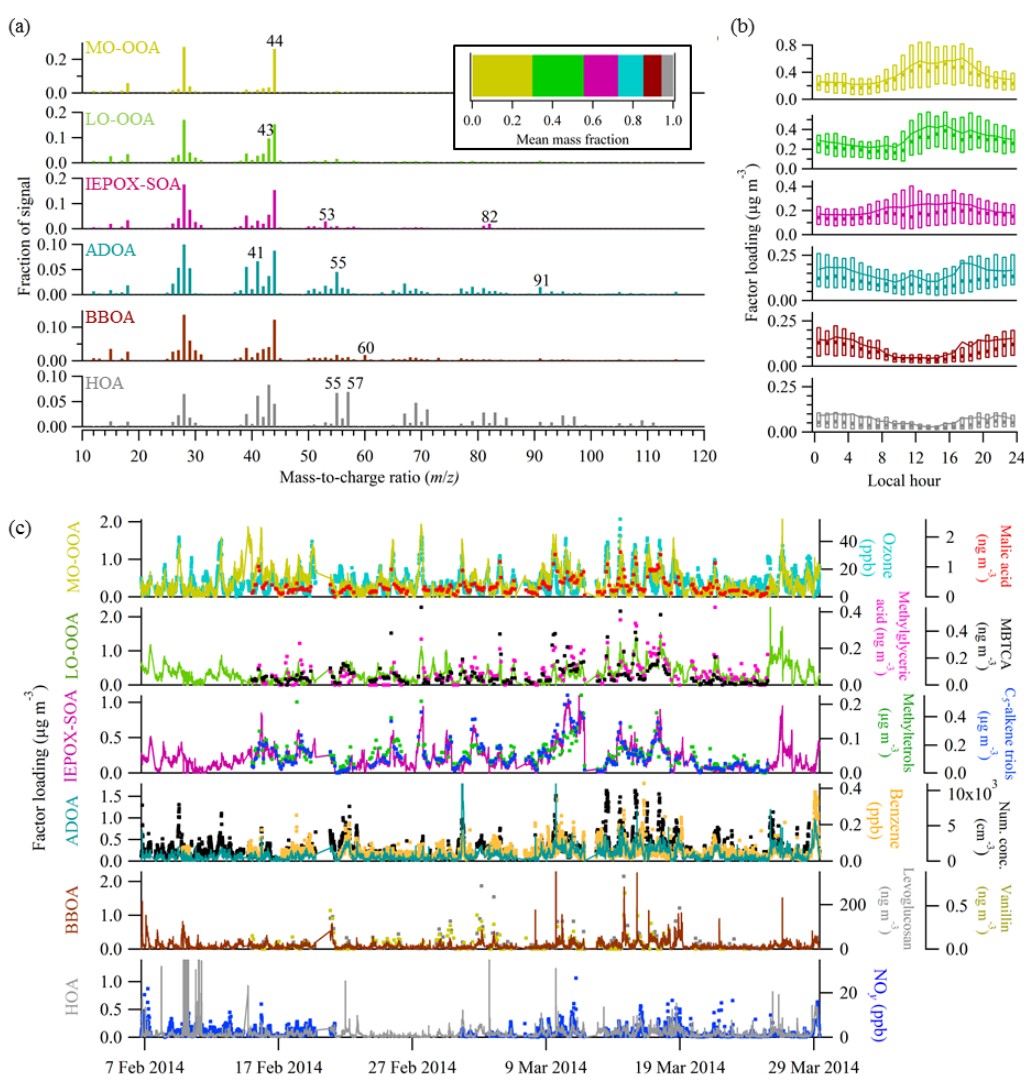

*Figure 4*



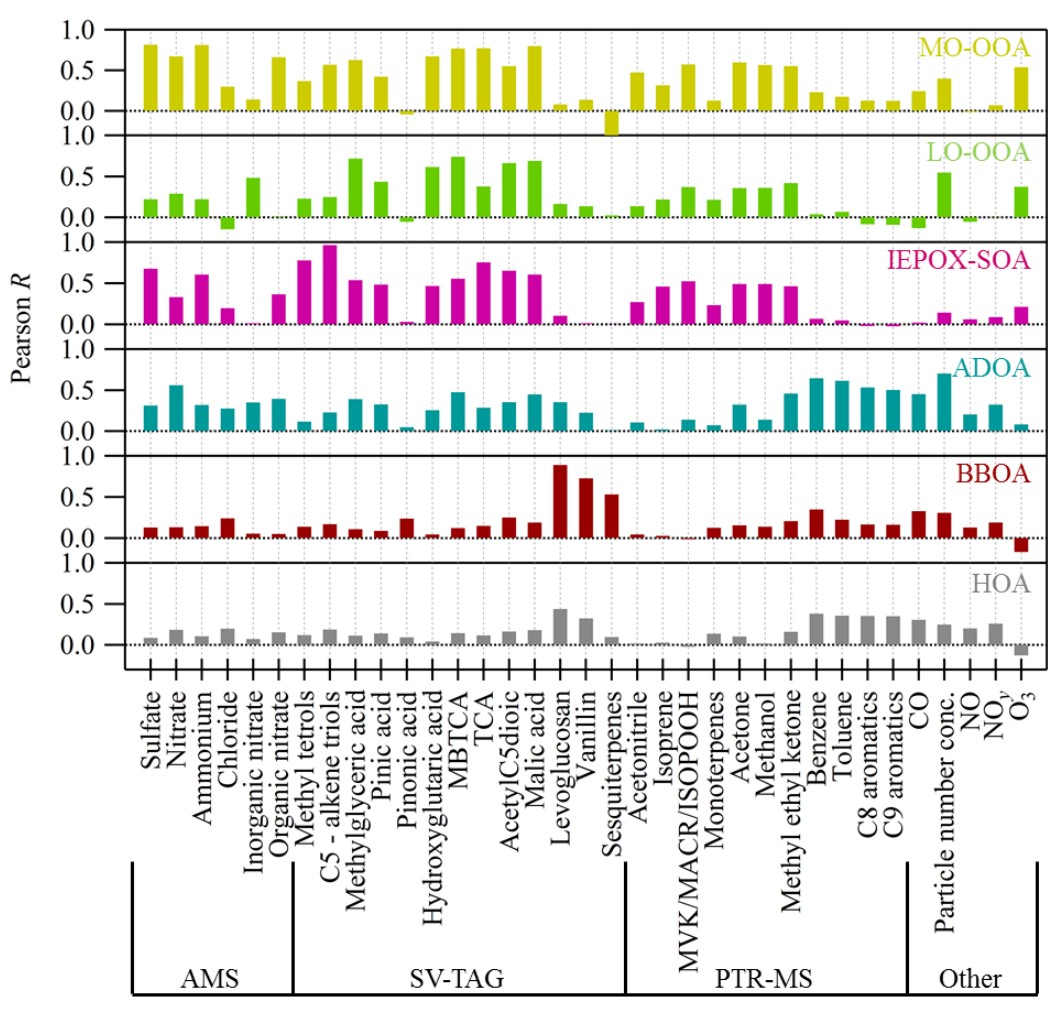

*Figure 5*



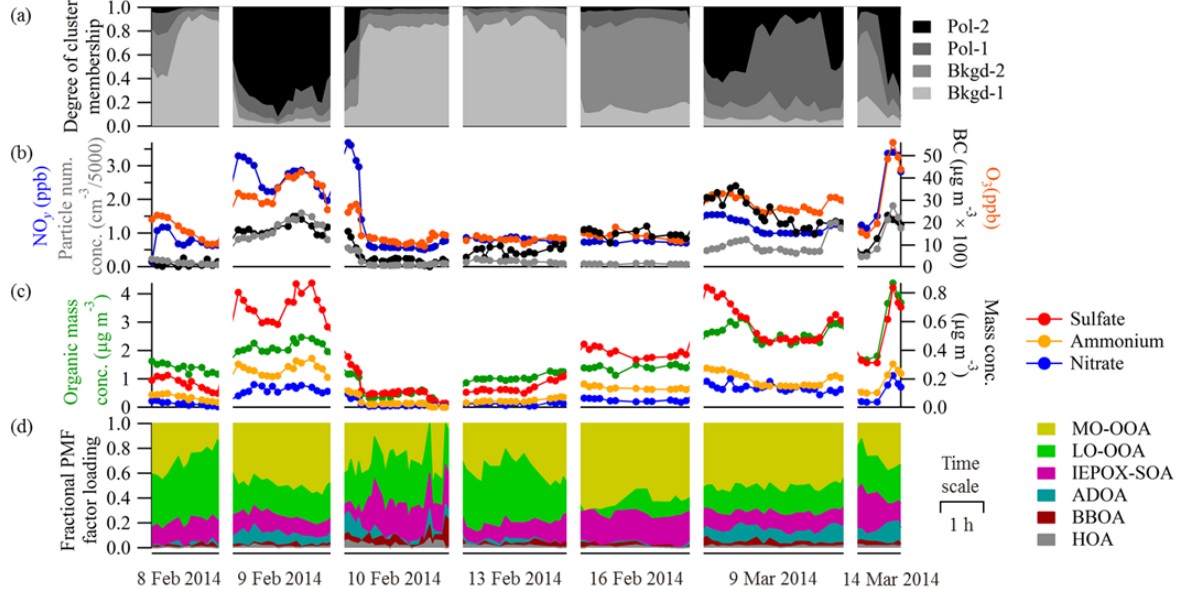

*Figure 6*





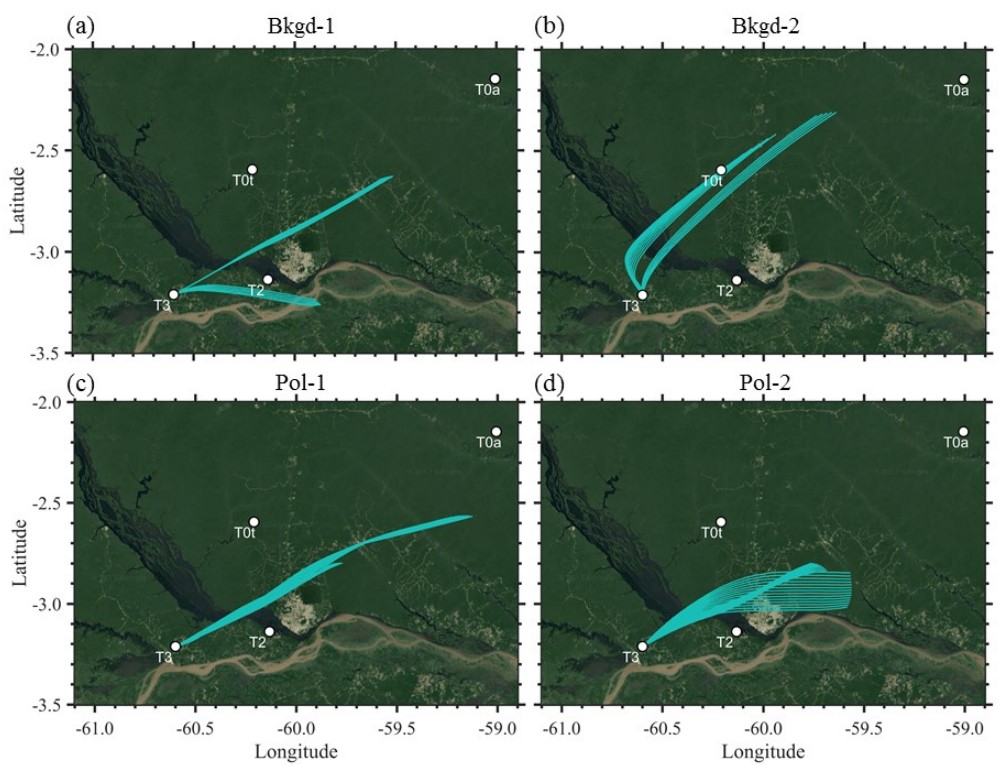





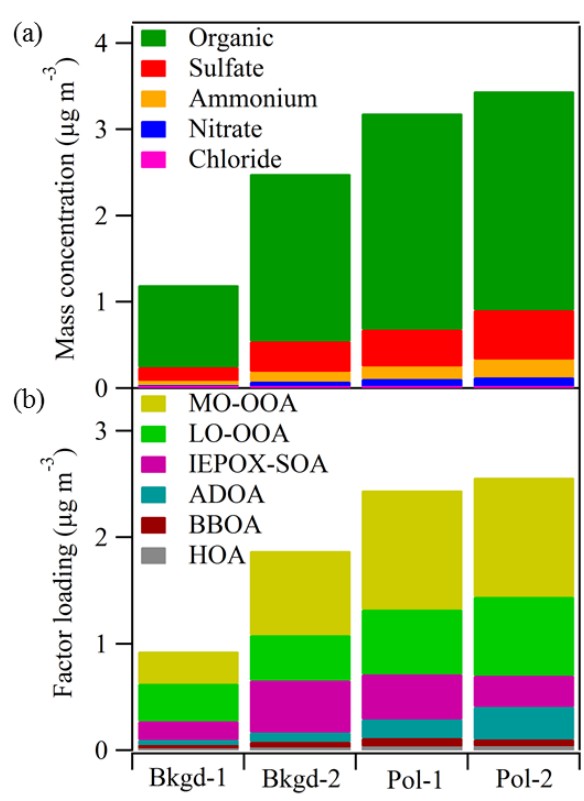

*Figure 8*



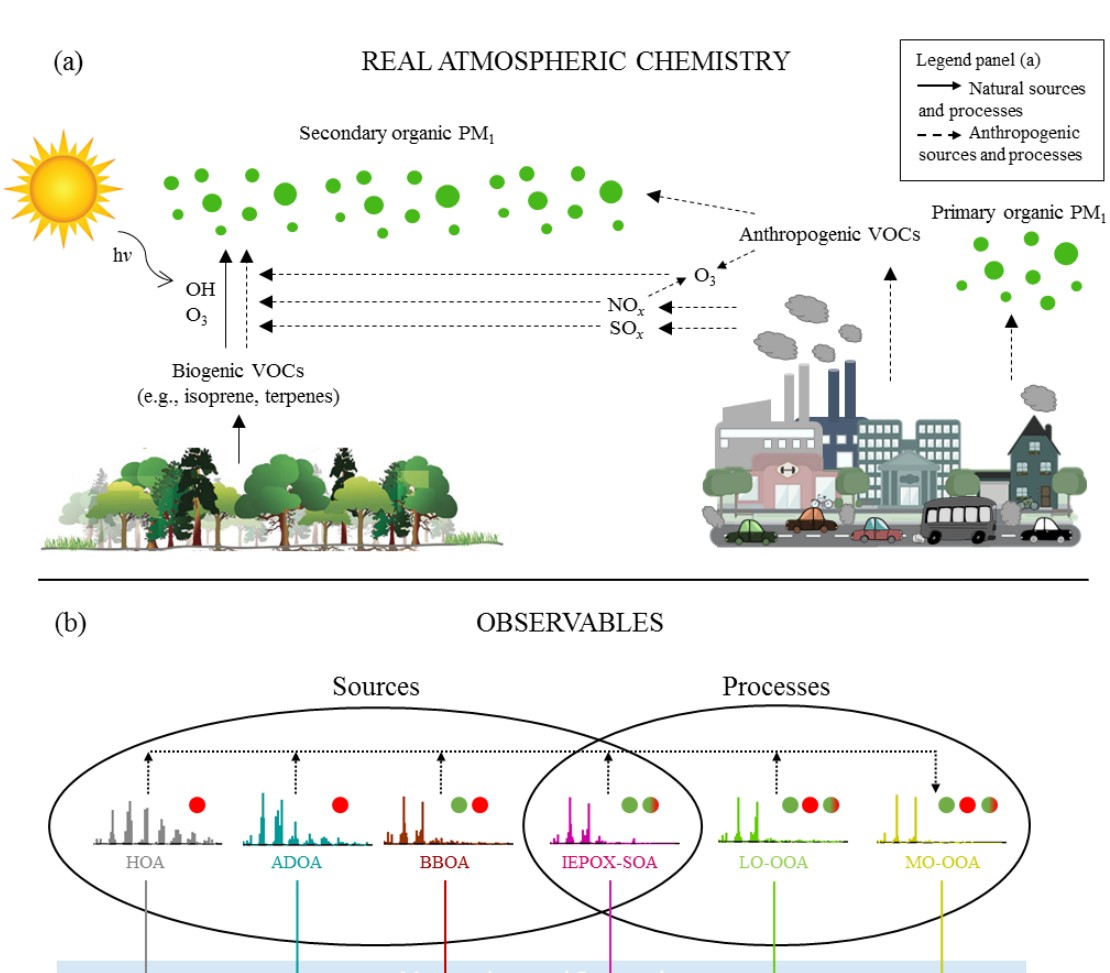

*Figure 9*