# Peer review of "Urban influence on the concentration and composition of submicron particulate matter in central Amazonia"

_Atmospheric Chemistry and Physics, 2018_

## Referee Comment (RC1) · Anonymous Referee #1 · 23 Apr 2018

The presented paper by de Sá et al, 'Urban influence on the concentration and composition of submicron particulate matter in central Amazonia' gives a very clear overview of the aerosol particle composition during the wet season in the Amazon region. The authors use two different methods to analyse AMS data. PMF, which gives an overview of the particle composition and fuzzy c-means algorithm to study the anthropogenic influence on the aerosol in Amazon.

I have few minor comments which are addressed in the following:

1. in the Introduction line 30, information on isoprene emissions compared to other biogenic or even anthropogenic VOCs could be added. Eg., how much isoprene is

estimated to be emitted globally, how much of it is emitted by the amazon rainforest?

2. Also in the Introduction,, in line with the measurement period that you are describing here, how many days of data did you collect during the IOP1. How frequently was the site influenced by the Manaus pollution during the time period presented in the manuscript?

3 in Methodology you mention 'V' and 'W' mode data. Maybe this is common knowledge but in my opinion it is useful to add a short description of what that means at least in the supplementary material.

4. in Auxiliary measurements and datasets, l. 124: it would be nice also for the supplementary measurements to add information for what time period that data was taken and how much of data was collected during each set of measurements.

5. in Results and discussion, in line 180 it is important to mention here again that the measurements at T0 sites were taken in a different year. It helps the reader.

Fig.3: This Figure contains too many data points, most of the points are hidden. I suggest to split the Figure into few sub-Figures, which enclose different time periods of the day. That allows to see any temporal trend of the particle evolution and to distinguish better what is happening at the different sites.

Fig.5: This Figure is easier to read and more informative, if the variables on the x-axis are grouped according to their source (biogenic, anthropogenic, background, biomass burning) other than the instrument they were measured with.

Fig. 7: the airmass back-trajectories are more valuable if they are calculated as ensembles rather than single trajectories. Ensemble gives you a group of trajectories which are all equally likely.

---

## Referee Comment (RC2) · Anonymous Referee #2 · 16 May 2018

his manuscript provides an overview of particle mass and chemical composition, with a focus on organic species in the Amazon. The authors strive to understand the anthropogenic contribution(s) to mass and influence on speciation and approach this with 2 different statistical approaches applied to online measurements. The work is interesting but seems premature. The paper relies on and cites several manuscripts that are "in preparation" to justify some arguments and conclusions and this is problematic. For example, comparison among different data sets (and presented in Figure 1) includes data not previously published, nor fully explained here.: data from ATTO sampling location "T0a-2015" is from Carbone et al., in preparation and "T2-2014" is from IOP1 Brito in preparation. Organic mass variability in relation to meteorology seems to be

an important finding and necessary to the arguments in this manuscript but Cirino et al. 'in prep' is the provided proof and readers are not left with sufficient information to understand the reasonableness of the argument.

The authors also cite de Sá et al 'in prep' to explain why a certain analysis is beyond the scope of this paper and not presented here and explain that the analysis is currently underway (e.g., biomass burning influence (presumably screened here?)) will be discussed in the literature later and I think that is ok.

Many journals would not even accept 'in prep' References at all. To use such References for conclusions seems unreasonable to me. Prior to acceptance for publication I think the 'prior' work must first be published or properly backed up here.

specific comments: Line 19/20: The choice to cite Weber et al., 2007 and Goldstein et al., 2009 here is curious. Weber et al. state in that paper: "Although NOx may be another precursor that could be influencing this system, NOx-WSOC . . . was weakly correlated" The R2 is <0.2 I acknowledge time scales for complex chemistry matter and correlation for instantaneous values can be low even though there is a dependence, however the work by Weber does not provide support for NOx or SO2 dependence as suggested by the authors here. The work by Weber et al does demonstrate a link to CO. The Goldstein analysis for particles is limited to satellite-AOT. Seasonal and spatial patterns have found these AOT observations are not due to organic fine particle mass (Ford and Heald, ACP 2013; Nguyen et al. GRL, 2016) The authors cite Xu et al., 2015 later in the manuscript and that would be a good citation here. Because the authors are talking in the manuscript here about the Southeast US, citing recent findings from the Southeast field campaigns (e.g., SOAS as in the Xu paper) and making a link with the context of those field campaigns would improve the paper.

I have no idea what "V" and "W" mode mean. The authors should provide an explanation if the distinction is important as they suggest.

Line 165: When talking about Figure 1 the authors state 'concentrations at the T2 site

[Figure]

were more than three times higher on average' All of the presented averages in Figure 1b overlap within the uncertainty. Can it be stated that there is statistically significance to the difference? Figure 2 suggests a factor of 2, not 3.

Figure 2 caption: Please correct the text: "Error! Reference source not found." The panels of Figure 2 have different y-axes and this should be mentioned explicitly.

Figure 4 is nice, but it's hard to read and digest.

Figure 9 is excellent!
* * *

---

## Author Comment (AC1) · 5 Jul 2018

The comment was uploaded in the form of a supplement:
https://www.atmos-chem-phys-discuss.net/acp-2018-172/acp-2018-172-AC1-supplement.zip

---

## Author Response (AR1)

**Response to reviews**

Reviewer comments are in **bold**. Author responses are in plain text. Excerpts from the manuscript are in *italics*. Modifications to the manuscript are in *blue italics*. Page and line numbers in the responses correspond to those in the ACPD paper.

Review #1

**The presented paper by de Sá et al, 'Urban influence on the concentration and composition of submicron particulate matter in central Amazonia' gives a very clear overview of the aerosol particle composition during the wet season in the Amazon region. The authors use two different methods to analyse AMS data. PMF, which gives an overview of the particle composition and fuzzy c-means algorithm to study the anthropogenic influence on the aerosol in Amazon. I have few minor comments which are addressed in the following:**

We thank the reviewer for the input, and the revised manuscript takes into account the comments and questions, as detailed in the responses below.

**1. in the Introduction line 30, information on isoprene emissions compared to other biogenic or even anthropogenic VOCs could be added. Eg., how much isoprene is estimated to be emitted globally, how much of it is emitted by the amazon rainforest?**

We thank the reviewer for this suggestion. Information on the importance of isoprene emissions, especially in the Amazon, is added to the revised manuscript, as follows:

Line 36:
*For tropical forests, isoprene emissions are especially important in PM production (Martin et al., 2010a; Chen et al., 2015Isoprene accounts for half of global BVOC mass emissions, and tropical forests are responsible for about 80% of terpenoid emissions (Guenther et al., 2012). In the Amazon, isoprene is the dominant BVOC emitted by vegetation and is estimated to contribute to about half of the organic PM concentrations under background conditions (Kuhn et al., 2010; Chen et al., 2015; Yáñez-Serrano et al., 2015) (Kuhn et al., 2010; Chen et al., 2015; Yáñez-Serrano et al., 2015).*

**2.a Also in the Introduction,, in line with the measurement period that you are describing here, how many days of data did you collect during the IOP1.**

The whole duration of IOP1 (Feb 1 to Mar 31, 2014) was the nominal operation time for all instruments including the AMS. The AMS data coverage is shown in Figure 1. The referred sentence in the introduction is clarified, as follows:

Line 60:

*The analysis employs data sets collected* *in the wet season from February 1 to March 31, 2014, corresponding to* *the first Intensive Operating Period (IOP1) of the GoAmazon2014/5 experiment (Martin et al., 2016).* *, corresponding to the wet season during the period of February 1 to March 31, 2014.*

**2.b How frequently was the site influenced by the Manaus pollution during the time period presented in the manuscript?**

We appreciate the suggestion, and the revised text includes this information as follows.

Line 72:

*The site was situated in a pasture of 2.5 km × 2 km surrounded by forest.* *Based on modeled flow trajectories of the pollution plume, the T3 site intercepted the plume about 40% of the time (Martin et al., 2017).*

**3. in Methodology you mention 'V' and 'W' mode data. Maybe this is common knowledge but in my opinion it is useful to add a short description of what that means at least in the supplementary material.**

The text is adjusted to restrict the use of these technical terms to the Supplementary Material, and an explanation of what these modes mean is included there.

Line 95:

*Organic, sulfate, ammonium, nitrate, and chloride PM mass concentrations were* *obtained from "V-mode" data. The choice of ions to fit was aided by "W-mode" data, which were collected for one of every five days.* *quantified.*

Line 208:

*Positive-matrix factorization was applied to the time series of the organic component of the high-resolution* *"V-mode"* *mass spectra (Ulbrich et al., 2009).*

Supplementary material, line 3:

*Quantification of mass concentrations by the AMS was obtained from "V-mode" data, which corresponds to the shorter ion time-of-flight path and is therefore the more sensitive mode. The choice of ions to fit was aided by "W-mode" data, which corresponds to the longer ion time-of-flight path and is therefore the mode with higher mass resolution. V-mode data were collected continuously, and W-mode data were collected for one of every five days.* *The time series of organic mass spectra measured by the AMS* *in V-mode* *was analyzed by positive-matrix factorization (PMF) using a standard analysis toolkit (Ulbrich et al., 2009).* *High-resolution "V-mode" data were used.*

**4. in Auxiliary measurements and datasets, l. 124: it would be nice also for the supplementary measurements to add information for what time period that data was taken and how much of data was collected during each set of measurements.**

Following this suggestion, the text is improved as follows.

Line 104:
*In complement to the AMS data set, the analysis herein incorporated auxiliary gas and particle measurements  collected during IOP1 at T3 (Martin et al., 2016).*

Line 124:
*At T2, non-refractory particle composition and concentration were measured by an Aerosol Chemical Speciation Monitor (ACSM) during the wet season from March 9 to April 30, 2014 (Cirino et al., submitted). ACSM measurements were  made at T0a during the wet season of 2015 , from February 1 to March 31 (Andreae et al., 2015). Further AMS datasets collected  at T0t during the wet season of 2008 (February 6 to March 22; AMAZE-08 campaign) were used in the analysis (Chen et al., 2009;Schneider et al., 2011). *

**5. in Results and discussion, in line 180 it is important to mention here again that the measurements at T0 sites were taken in a different year. It helps the reader.**

The reviewer raises a good point, and the text is adjusted as follows. (The reviewer mentioned line 180, but the information about T0 was at line 163, and we think this is what the reviewer meant.)

Line 163:
*The NR-PM$_1$ mass concentrations at the T0 sites upwind of Manaus, although measured in different years, were consistently around  1 µg m$^{-3}$.*

**6. Fig.3: This Figure contains too many data points, most of the points are hidden. I suggest to split the Figure into few sub-Figures, which enclose different time periods of the day. That allows to see any temporal trend of the particle evolution and to distinguish better what is happening at the different sites.**

Based on this feedback, the figure is revised to provide a better visualization of the data points. The caption of Figure 3 is adjusted as follows:

*Gray and blue circles correspond, respectively, to measurements at T3 and T2 during IOP1, in the wet season of 2014. For visualization purposes, the two datasets are plotted separately in panels a and b.*

The intention of the authors for this figure is to provide a general comparison of the PM oxidation at both sites. For the reviewer's suggestion in relation to plume evolution, we think that a more elaborate analysis beyond the scope of the authors' intention would be necessary to take into account (i) the transport time between the sites for each individual data point, (ii) whether the plume passed over both sites, and (ii) meteorological factors. Cirino et al., submitted focused on this kind of complex plume analysis. The text is adjusted to reflect these important points.

Line 198:
*The comparison depicted in Figure 3*  *indicates the effects of the plume over the 4 h of transport from T2 to T3, which were investigated in detail by Cirino et al. (submitted).*

**7. Fig.5: This Figure is easier to read and more informative, if the variables on the x-axis are grouped according to their source (biogenic, anthropogenic, background, biomass burning) other than the instrument they were measured with.**

We appreciate this suggestion. Based on it, the authors prepared both possible figures for internal discussion. In the end, the authors believe that the original figure is better for presentation. The reason is that most of the tracers have contributions from several different and in some cases unknown sources so that a definitive classification would be uncertain, and the revised figure would be a scientific over-stretch. The exceptions, such as levoglucosan which is a specific tracer for biomass burning, are explicitly mentioned in the text.

**8. Fig. 7: the airmass back-trajectories are more valuable if they are calculated as ensembles rather than single trajectories. Ensemble gives you a group of trajectories which are all equally likely.**

We thank the reviewer for this thoughtful comment, which generated significant internal discussion among the authors. Although the idea is appreciated, the use of ensembles did not seem the most appropriate or necessary tool for the analysis of Figure 7.

The backtrajectories are employed in this study in a supportive rather than central role to the clustering analysis, i.e., trajectories are not used to generate clusters but rather to help in their interpretation. In each panel of Fig. 7, trajectories are representative of the case studies shown in Fig. 6. It is visually clear that trajectories within those time periods are already clustered. Calculating ensembles for all observation times (every 12 min) over the course of the two months of the study period would have added a very large computational time and human expense that could not be afforded. Hence, the cost-benefit of a more complex trajectory analysis was not justified, while the single trajectories still added value to the data interpretation.

Review #2

**1. [T]his manuscript provides an overview of particle mass and chemical composition, with a focus on organic species in the Amazon. The authors strive to understand the anthropogenic contribution(s) to mass and influence on speciation and approach this with 2 different statistical approaches applied to online measurements. The work is interesting but seems premature. The paper relies on and cites several manuscripts that are "in preparation" to justify some arguments and conclusions and this is problematic. For example, comparison among different data sets (and presented in Figure 1) includes data not previously published, nor fully explained here.: data from ATTO sampling location "T0a-2015" is from Carbone et al., in preparation and "T2-2014" is from IOP1 Brito in preparation.**

We thank the reviewer for reading the manuscript and providing valuable feedback.

For the references "in preparation", the following revisions are made.
- (i)     Carbone et al. (in preparation) is replaced by Andreae et al. (2015), which already published the T0a-2015 data used in this study.
- (ii)    Brito et al. (in preparation) is replaced by Cirino et al. (submitted). This manuscript is in the final stages of peer review, and we expect that "submitted" can be replaced with a full citation for the ACP publication of the present manuscript. Both Brito and Cirino, responsible for ACSM data collection and analysis at T2, are co-authors in the present study.

For changes made to the text to address (i) and (ii) please see reply #4 to review #1. The references in the caption of Figure 1 are also accordingly updated.

We believe that these updates satisfy the reviewer's concern about "premature". Importantly, the comparison of the T3 composition to other sites only appears in one section of the manuscript (lines 155-205). The main conclusions, by contrast, largely come from the combined analysis of PMF and FCM on the data collected at T3, which is completely original and presented in detail in the following sections of the manuscript (lines 206-514).

**2. Organic mass variability in relation to meteorology seems to be an important finding and necessary to the arguments in this manuscript but Cirino et al. 'in prep' is the provided proof and readers are not left with sufficient information to understand the reasonableness of the argument.**

This reference was adjusted as follows:

Line 174:
*This influence waxes and wanes with small northerly or southerly shifts of the trade winds as well as other changes in regional circulation tied to daily meteorology (* *preparationdos Santos et al., 2014; Martin et al., 2017).*

**3. The authors also cite de Sá et al 'in prep' to explain why a certain analysis is beyond the scope of this paper and not presented here and explain that the analysis is currently underway (e.g., biomass burning influence (presumably screened here?)) will be discussed in the literature later and I think that is ok.**

The current manuscript focuses on the wet season. As stated in the introduction (line 48), the influence of biomass burning is minimal during the wet season. The reference to de Sá et al. (in preparation) in the introduction (line 64) was intended to highlight that there is a separate manuscript under way for the dry season (to also be submitted to the GoAmazon2014/5 Special Issue of ACP). Although related, these studies are independent.

**4. Many journals would not even accept 'in prep' References at all. To use such References for conclusions seems unreasonable to me. Prior to acceptance for publication I think the 'prior' work must first be published or properly backed up here.**

We understand the reviewer's concerns. Please see replies 10 and 11.

**5. specific comments: Line 19/20: The choice to cite Weber et al., 2007 and Goldstein et al., 2009 here is curious. Weber et al. state in that paper: "Although NOx may be another precursor that could be influencing this system, NOx-WSOC… was weakly correlated" The R2 is <0.2 I acknowledge time scales for complex chemistry matter and correlation for instantaneous values can be low even though there is a dependence, however the work by Weber does not provide support for NOx or SO2 dependence as suggested by the authors here. The work by Weber et al does demonstrate a link to CO. The Goldstein analysis for particles is limited to satellite-AOT. Seasonal and spatial patterns have found these AOT observations are not due to organic fine particle mass (Ford and Heald, ACP 2013; Nguyen et al. GRL, 2016) The authors cite Xu et al., 2015 later in the manuscript and that would be a good citation here. Because the authors are talking in the manuscript here about the Southeast US, citing recent findings from the Southeast field campaigns (e.g., SOAS as in the Xu paper) and making a link with the context of those field campaigns would improve the paper.**

We greatly appreciate that the reviewer pointed out this mismatch in citations. The text is updated as follows:

Line 15:
*In the northeastern USA, de Gouw et al. (2005) showed that*  *concentrations of organic particulate matter (PM) correlated well with anthropogenic tracers, yet the concentrations of the anthropogenic precursors were insufficient to explain the observed PM concentrations. In the southeastern USA, radioisotope analysis of organic PM determined that 70% to 80% of the carbon mass had a modern origin even as correlations were observed between SOM mass concentrations and anthropogenic VOC and CO concentrations (Weber et al., 2007). This finding and those of further field studies in the region together suggested that the organic PM was produced mainly from biogenic VOCs (BVOCs) yet modulated by anthropogenic emissions of NO$_x$ and SO$_2$ (Hu et al., 2015; Xu et al., 2015a; Xu et al., 2015b; Zhang et al., 2018).*

**6. I have no idea what "V" and "W" mode mean. The authors should provide an explanation if the distinction is important as they suggest.**

Please see reply #3 to review #1.

**7. Line 165: When talking about Figure 1 the authors state 'concentrations at the T2 site were more than three times higher on average' All of the presented averages in Figure 1b overlap within the uncertainty. Can it be stated that there is statistically significance to the difference? Figure 2 suggests a factor of 2, not 3.**

The reviewer raises an important point that needs clarification. Firstly, the bars in Figure 1b do not represent uncertainty but rather variability in the measurements. Secondly, the variability of concentrations among sites can only be fairly compared by considering different times of day as was done in Figure 2, since the variability is largely driven by the diel trends. To clarify this point to the reader, we removed the bars in Figure 1b and emphasized in the caption of Figure 1 as well as in the text that a comparison of the variability across sites is presented in Figure 2, as follows.

Figure 1 caption:
* The variability of measurements across sites is evaluated in Figure 2.*

Line 170:

*The diel trends of organic and sulfate mass concentrations as well as their variabilities across the four sites are shown in Figure 2.*

**8. Figure 2 caption: Please correct the text: "Error! Reference source not found." The panels of Figure 2 have different y-axes and this should be mentioned explicitly.**

We thank the reviewer for catching these two points, and corrections are made in the figure caption as follows:

*"... at four different sites (cf. Fig.1 and Fig.  S1). The ordinate scale for the T2-2014 panel is twice that of the other panels. Mass concentrations were corrected to standard temperature and pressure (273.15 K and 105 Pa). Local time is (UTC - 4 h). Lines represent means, solid markers show medians, and boxes span interquartile ranges. *

**9. Figure 4 is nice, but it's hard to read and digest.**

We understand that the information content of Figure 4 is high. This study heavily relies on the PMF results, and Figure 4 provides an important summary of the PMF factors. The authors discussed several alternative representations for this figure and believe that the present version is the best option. Importantly, the text is optimized to accompany the figure. Lines 228 - 349 of the text are paired to the reading of Figure 4. The text explains the characteristics of the factors one by one in each paragraph and systematically refers to each of the panels in that figure.

**10. Figure 9 is excellent!**

Thank you!

[revised manuscript text omitted]

Figure 2

[Figure]

*Figure 3*

[Figure]

*Figure 4*

[Figure]

*Figure 5*

[Figure]

*Figure 6*

[Figure]

Figure 7

[Figure]

*Figure 8*

[Figure]

(a) REAL ATMOSPHERIC CHEMISTRY

Legend panel (a)
→ Natural sources and processes
- - → Anthropogenic sources and processes

Secondary organic PM$_1$

Primary organic PM$_1$

Anthropogenic VOCs

$h\nu$

OH
O$_3$

O$_3$
NO$_x$
SO$_x$

Biogenic VOCs
(e.g., isoprene, terpenes)

[Figure]

(b) OBSERVABLES

Sources

Processes

HOA    ADOA    BBOA    IEPOX-SOA    LO-OOA    MO-OOA

Meteorology and Geography

Pol-2    Pol-1    Bkgd-2    Bkgd-1

Legend panel (b)
········→ Aging mechanisms
PM origin:
● biogenic
● anthropogenic
● bio-anthrop. interactions

Clusters: conditions at the receptor site

*Figure 9*

**Supplementary Material for**

**Urban influence on the concentration and composition of submicron particulate matter in central Amazonia**

Suzane S. de Sá (1), Brett B. Palm (2), Pedro Campuzano-Jost (2), Douglas A. Day (2), Weiwei Hu (2), Gabriel Isaacman-VanWertz[a] (3), Lindsay D. Yee (3), Joel Brito[b] (4), Samara Carbone[c] (4), Igor O. Ribeiro (5), Glauber G. Cirino[d] (6), Yingjun J. Liu[e] (1), Ryan Thalman[f] (7), Arthur Sedlacek (7), Aaron Funk (8), Courtney Schumacher (8), John E. Shilling (9), Johannes Schneider (10), Paulo Artaxo (4), Allen H. Goldstein (3), Rodrigo A.F. Souza (5), Jian Wang (7), Karena A. McKinney[g] (1), Henrique Barbosa (4), M. Lizabeth Alexander (11), Jose L. Jimenez (2), Scot T. Martin[*] (1, 12)

(1) School of Engineering and Applied Sciences, Harvard University, Cambridge, Massachusetts, USA
(2) Department of Chemistry and Cooperative Institute for Research in Environmental Sciences, University of Colorado, Boulder, Colorado, USA
(3) Department of Environmental Science, Policy, and Management, University of California, Berkeley, California, USA
(4) Institute of Physics, University of São Paulo, São Paulo, Brazil
(5) School of Technology, Amazonas State University, Manaus, Amazonas, Brazil
(6) National Institute for Amazonian Research, Manaus, Amazonas, Brazil
(7) Brookhaven National Laboratory, Upton, New York, USA
(8) Department of Atmospheric Sciences, Texas A&M University, College Station, Texas, USA
(9) Atmospheric Sciences and Global Change Division, Pacific Northwest National Laboratory, Richland, WA, USA
(10) Particle Chemistry Department, Max Planck Institute for Chemistry, Mainz, Germany
(11) Environmental Molecular Sciences Laboratory, Pacific Northwest National Laboratory, Richland, Washington, USA
(12) Department of Earth and Planetary Sciences, Harvard University, Cambridge, Massachusetts, USA
[a] Now at Department of Civil and Environmental Engineering, Virginia Tech, Blacksburg, Virginia, USA
[b] Now at Laboratory for Meteorological Physics (LaMP), University Blaise Pascal, Aubière, France
[c] Now at Federal University of Uberlândia, Uberlândia, Minas Gerais, Brazil
[d] Now at Department of Meteorology, Geosciences Institute, Federal University of Pará, Belém, Brazil
[e] Now at University of California, Berkeley, California, USA
[f] Now at Department of Chemistry, Snow College, Richfield, Utah, USA
[g] Now at Colby College, Waterville, Maine, USA

Submitted: February 2018

*Atmospheric Chemistry and Physics*

[*]To Whom Correspondence Should be Addressed

*E-mail: scot_martin@harvard.edu*

*https://martin.seas.harvard.edu/*

**S1. Positive-matrix factorization**

**S1.1 Diagnostics of the six-factor solution**

Quantification of mass concentrations by the AMS was obtained from "V-mode" data, which corresponds to the shorter ion time-of-flight path and is therefore the more sensitive mode. The choice of ions to fit was aided by "W-mode" data, which correspond to the longer ion time-of-flight path and is therefore the mode with highest mass resolution. V-mode data were collected continuously, and W-mode data were collected for one of every five days. The time series of organic mass spectra measured by the AMS in V-mode was analyzed by positive-matrix factorization (PMF) using a standard analysis toolkit (Ulbrich et al., 2009). High-resolution "V-mode" data were used. The PMF solution was based on minimization of the "Q-value" (i.e., the sum of the weighed squared residuals for a chosen number of factors) and the physical meaningfulness of factors, as evaluated by profile characteristics and correlations with gas and particle phase measurements by other instruments.

Technical diagnostics of the six-factor solution are presented in Figure S3 in complement to the diagnostics presented in de Sá et al. (2017). The analysis was run for a number of factors from 1 to 10, and the rotational ambiguity parameter $f_{peak}$ was varied from -1 to 1 in intervals of 0.2. Panel a shows the statistics of residuals for solutions with different number of factors. There was a large improvement in the solution when a sixth factor was introduced, as shown by a significant decrease in residuals, and only a marginal improvement when a seventh factor was added. Panel b shows, on the ordinate, the correlation between the time series of loadings for each pair of factors and, on the abscissa, the correlation between the profiles of each pair of factors. For the six-factor solution, the correlations among factor profiles are overall lower, also suggesting a better separation of factors and an improvement in the solution. Figure S4

corroborates this analysis by showing the factor profiles and loading time series of the 5- and 7- factor solutions. In the 5-factor solution, factors 4 and 5 seem to be a result of mixing of the three factors that are associated with secondary processing in the 6-factor solution (MO-OOA, LO-

OOA, IEPOX-SOA). Conversely, in the 7-factor solution, some splitting seems to occur as factor

7 is physically meaningless, and a few pairs of factors have higher correlations between their loading time series (cf. Figure S3). An $f_{peak}$ of zero was chosen for the final 6-factor solution, since it yielded the minimum quality of fit parameter $Q/Q_{expected}$ de Sá et al. (2017), and no significant improvements in the external validation of factors were observed by varying $f_{peak}$.

**S1.2 Discussion of the ADOA PMF factor**

 ADOA is interpreted as a primary anthropogenic factor due to the correlation of its loadings with several tracers of anthropogenic activities (Figure 5), its spectral profile, and its diel behavior (Figure 4). Even though factors containing a characteristic $m/z$ 91 have been reported in the literature as a biogenic factor (Robinson et al., 2011; Budisulistiorini et al., 2015;

Chen et al., 2015; Riva et al., 2016), the ADOA of this study showed similarity with primary organic material from cooking activities. Figure S5 shows the high similarity of ADOA of this study to a factor representing cooking emissions at an urban background site in Barcelona, Spain (Mohr et al., 2012), and to a factor representing a cooking source tied to restaurants in an urban background site in Zurich, Switzerland (Lanz et al., 2007). By contrast, a lower similarity is found with the "91fac" factor found in the Borneo forest, a predominantly biogenic site. This result emphasizes that a characteristic marker ion $C_7H_7^+$ at $m/z$ 91 does not directly imply either biogenic or anthropogenic origin, and the interpretation of a PMF factor with such marker should also strongly rely on the atmospheric context of the measurements, including the correlations of the factor loadings with external measurements and the diel behavior.

**S2. Estimates of organic and inorganic nitrates based on AMS analysis**

The typical AMS analysis reports total nitrate, meaning that nitrate fragments originating from both organic and inorganic nitrates are reported indistinctively as nitrate. In the absence of external measurements of inorganic nitrate, an estimation method using the ratio of $NO_2^+$ to $NO^+$ signal intensities measured by the AMS was employed (Figure S6; Fry et al., 2009; Farmer et al., 2010; Fry et al., 2013). Calculations were done on a 60-min time base to increase signal over noise. The obtained organic and inorganic nitrate time series were then interpolated into the original AMS timestamp for ambient measurements (i.e., one point every 8-min interval). The analysis excluded points that had total nitrate below the estimated detection limit, $DL_{Nitrate}$, which was estimated as three times the standard deviation for "closed AMS spectra", i.e., when chopper was in closed position and particles did not reach the vaporizer. Mathematically,

$DL_{Nitrate} = 3 \times \sqrt{E}$, where $E$ is the "closed" error calculated by the standard *PIKA* software (Ulbrich et al., 2009). The dark blue dashed line in Figure S6c that defines $NO_2^+/NO^+$ for inorganic nitrate was determined by linear fit of ammonium nitrate calibrations performed regularly, as shown by the grey triangles. The small drift over time can be attributed to a gradual clean-up of the vaporizer. Worth noting, whether the linear fit or an average value was used for the calculations, the overall results did not change considerably, as all calibration ratios lied within ± 20% of the campaign-average ratio. The ratio $NO_2^+/NO^+$ for organic nitrates was assumed to be a factor of 2.25 lower than that of inorganic nitrate based on previous field studies (Farmer et al., 2010; Fry et al., 2013). The resulting IOP1-average for the fraction of organic nitrate in total nitrate (Figure S6b) was 87%.

**S3. Fuzzy c-means clustering**

69   Fuzzy c-means (FCM) clustering was applied to the dataset consisting of concentrations

70 of particle number, $NO_y$, ozone, black carbon, and sulfate (Bezdek et al., 1984). The use of a

71 fuzzy clustering method stems from the understanding that any point in time may be affected by

72 a combination of different sources and processes and could therefore be anywhere on the scale

73 between pristine background and extreme polluted conditions, as opposed to a simpler binary

74 classification. Given the scope of the analysis as non-overcast afternoon times, data points were

75 restricted to (i) local 12:00-16:00 h, (ii) local solar radiation over the past 4 h not less than 200 W

76 $m^{-2}$ (i.e., excluding the lower 20 percentile), and (iii) insignificant precipitation ($< 0.1$ mm) over

77 the previous 10 h along backward trajectory (a threshold was used as most rain radar grid cells

78 had non-zero yet negligible values). The data were normalized prior to the FCM analysis using

79 the z-score method, which transforms all variables into a common scale with an average of zero

80 and standard deviation of one.

81   The FCM algorithm minimizes the objective function represented in Eq. S1, which is a

82 weighted sum of squared errors where the error is the Euclidean distance between each data

83 point and a cluster centroid.

84        $$J(U,v)= \sum_{k=1}^{N} \sum_{i=1}^{c} u_{ik}{}^{m} \left\| y_k - v_i \right\|^2$$       (Eq. S1)

85 The input data is given by the matrix $Y = [y_1, y_2, …, y_N]$, where $y_k$ is a vector of length $X$ at the $k$-

86 $th$ time point. $X$ is the number of variables (i.e., measurements) used as input in the analysis. The

87 number of time points is represented by $N$, and the associated running index is $k$. $N$ in this case

88 was 313. The number of clusters is represented by $c$, and the corresponding running index is $i$.

89 The coordinates of the centroid of each cluster $i$ are represented by $v_i$, a vector of length $X$. The

90 exponent of the Fuzzy partition matrix is represented by $m$. The algorithm returns (1) the Fuzzy

91 partition matrix of $Y$, given by $U = [u_{ik}]$ where $u_{ik}$ is the degree of membership of time point $k$ to cluster $i$, (2) the vectors of coordinates of cluster centers, given by $v = [v_i]$, as well as (3) the value $J$ of the objective function.

The analysis was performed in MATLAB® using the "fcm" function in the Fuzzy logic toolbox$^{TM}$. The stop criterion of the algorithm is that either the maximum number of iterations is reached or the improvement of the objective function between two consecutive iterations is less than the minimum amount of improvement specified. The default value of $1 \times 10^{-5}$ was used for the minimum amount of improvement, and the maximum number of iterations was set to 1000 so that convergence always happened before this maximum was reached. A default value of 2 was used for the exponent $m$ of the partition matrix. Fuzzy clustering algorithms are not sensitive to small fluctuations in $m$ (Chatzis, 2011), and a value in the range of 1.5 to 3 is recommended (Bezdek et al., 1984; Hathaway and Bezdek, 2001).

The analysis was run for a number of clusters varying from two to eight, and the value of the objective function for each run is shown in Figure S7. The choice of number of clusters hinges on a balance between increased complexity and additional information provided by each extra cluster. The improvement in the objective function was larger in the range of two to four clusters, with marginal improvements above four clusters (Figure S7).  The location of cluster centroids was also examined for evaluation of cluster overlap (Figure S8). The addition of a fifth cluster made two pairs of clusters very similar, as can be seen by the locations of cluster centroids in Figure S8. The solution of four clusters was therefore a reasonable choice to represent the studied system. The subsequent characterization of the PM chemical composition associated with each cluster further confirmed the meaningfulness of the solution. Although the three-cluster solution could also provide a reasonable representation of the system, the four- cluster solution provided further insight by differentiating two background and two polluted conditions.

Subsequently, the PM composition associated with each of the clusters was determined by calculating the corresponding coordinates of the centroids for AMS species concentrations and PMF factor loadings, which were not input to the FCM analysis. The calculation followed the mathematical definition of the centroid (Eq. S2). The resulting characterization of clusters is shown in Figure 8 and Table 2.

                        $$v_i = \frac{\sum_{k=1}^{N} (u_{ik})^m y_k}{\sum_{k=1}^{N} (u_{ik})^m}$$                (Eq. S2)

**List of Supplementary Figures**

**Figure S1.** Location of the GoAmazon2014/5 sites relevant for this study. Image data: Google earth.

**Figure S2.** Scatter plot of the AMS signal fraction at $m/z$ 44 ($f_{44}$) against that at $m/z$ 43 ($f_{43}$). Green and yellow markers correspond to measurements made by two different AMS instruments at T0t in the wet season of 2008 during the AMAZE-08 campaign (Chen et al., 2009; Schneider et al., 2011). Red markers correspond to measurements made at the T0a (ATTO) by an ACSM during the wet season of 2015. A correction factor of 0.75 was applied to the $f_{44}$ values of the ACSM based on calibrations with standards. Solid squares represent median values, and whiskers represent 10 and 90 percentiles. The plot shows a significant variability between the observations of 2008 and 2015 for the two background sites. An explanation of the differences is not attempted herein and warrants further investigation through longer-term continuous measurements.

**Figure S3.** Diagnostics of the PMF analysis. (a) Statistics of the sum of for solutions with different number of factors. Box plots show the interquartile ranges, including the medians as a horizontal line. Red markers show the means. Whiskers show the 5 and 95 percentiles. (b) Correlations expressed as between each pair of factors within each PMF solution, with number of factors varying from 2 to 7. The Pearson $R$ value between factor loadings is shown on the coordinate and between factor profiles is shown on the abscissa. Numbers in red indicate the identity of the pair of factors.

**Figure S4.** Results of the PMF analysis for 5 factors (a and b) and 7 factors (c and d). Panels on the left (a and c) show the time series of factor loadings and panels on the right (b and d) show the profiles of factors. The signals shown in panels b and d were summed to unit mass resolution.

**Figure S5.** Comparison of the ADOA factor profile from the present study to factors found in three other field studies. "COA" are factors representative of cooking activities, and the "91fac" from Robinson et al. (2011) was tied to biogenic sources.

**Figure S6.** Summary of the analysis for estimating organic and inorganic nitrates from AMS bulk measurements. (a) Resulting time series of organic and inorganic nitrates are shown together with the original nitrate AMS times series. (b) Time series of the fraction of organic nitrate in total nitrate. (c) Time series of the measured $NO_2^+/NO^+$ ratio is shown in red and values of $NO_2^+/NO^+$ from ammonium nitrate calibrations are shown in gray triangles. A linear fit to those calibration ratios is shown by the dashed dark blue line and constitutes the reference ratio for inorganic nitrate over time. The dashed light blue line is the reference ratio for organic nitrates over time. Calculations were done for data binned to one hour (as plotted), and the resulting time series were interpolated to the native time stamp for evaluation of correlations in the PMF analysis.

**Figure S7.** Value of the objective function of the FCM analysis (Eq. 1) in the last iteration plotted against the number of clusters.

**Figure S8.** Locations of cluster centroids from the FCM analysis as visualized by a 2-D projection on the plane defined by each pair of input variables. Results for two to five clusters are shown in panels a to d. Red circles are observational data and black squares are cluster centroids.

**Figure S9.** Map of Manaus city depicting population density as well as main avenues and representative locations of industry, restaurants, and other businesses. Population density data are from the 2010 census by the Brazilian Institute of Geography and Statistics (IBGE, 2010).

**Figure S10**. Measurements showing the geographical heterogeneity of emissions from Manaus. On the top row, concentrations of sulfate (red) and particle number (white) measured onboard the G-1 aircraft on (a) March 19 and (b) Mar 21. Image data: Google earth. On the bottom row, rose plots of mean (c) sulfate mass concentrations and (d) particle number concentrations observed at T2 during IOP1. The angles represent wind direction, the radial scale (0 to 5 m s$^{-1}$) represents wind speed, and the color scale represents the concentrations. The interactions of emissions from Manaus with the daily river breeze is complex, and the detailed interpretation of the data sets is not fully attempted herein. Of importance, the river breeze terminates well below 500 m based on the G-1 flights so that the complexities of the river breeze largely do not affect the measurements at T3 because most pollution is lofted above the river breeze before reaching T3 (Medeiros et al., in preparation). These surface-level plots, although complicated by the river breeze, demonstrate the heterogeneity of Manaus emissions.

[Figure]

*Figure S1*

[Figure]

*Figure S2*

[Figure]

*Figure S3*

[Figure]

*Figure S4*

[Figure]

*Figure S5*

[Figure]

*Figure S6*

[Figure]

*Figure S7*

[Figure]

*Figure S8*

[Figure]

*Figure S9*

[Figure]

(a) 19 Mar 2014

(b) 21 Mar 2014

(c) IOP1, T2

(d) IOP1, T2

Figure S10

---

## Referee Report (RR1)

Referee comment on 'Urban influence on the concentration and composition of submicron particulate matter in central Amazonia'.

I thank the authors for updating the manuscript, I think it has improved a lot and I only have a few additional comments.

In line 67, the authors state: A separate publication is planned for IOP2, 67 corresponding to the dry season period of August 15 to October 15, when biomass burning 68 emissions were prevalent (de Sá et al., in preparation).

I think that this sentence here is not needed as it doesn't add any additional valuable information to the presented manuscript.

The Figures are really nice and the presentation of the results is very well done. I am missing a proper description of Figure 5 in the text though. Even a few sentences with the key points of what we see in Figure 5 would be very useful already.

My other comment is related to Figure 4: I agree with the other referee that it contains a lot of information and is hard to digest, but I think the authors are giving very good guidance through it in the text. I am not fully convinced that panel (c) in Figure 4 is really necessary. I am not sure how much it adds to the information already presented in the other panels and in Figure 5. I understand if the authors want to keep it as it is also.

---

## Author Response (AR2)

**Response to Report #2**

Reviewer comments are in **bold**. Author responses are in plain text. Excerpts from the manuscript are in *italics*. Modifications to the manuscript are in *blue italics*.

**I thank the authors for updating the manuscript, I think it has improved a lot and I only have a few additional comments.**
Thank you.

**In line 67, the authors state: A separate publication is planned for IOP2, 67 corresponding to the dry season period of August 15 to October 15, when biomass burning 68 emissions were prevalent (de Sá et al., in preparation). I think that this sentence here is not needed as it doesn't add any additional valuable information to the presented manuscript.**
This sentence was removed.

**The Figures are really nice and the presentation of the results is very well done. I am missing a proper description of Figure 5 in the text though. Even a few sentences with the key points of what we see in Figure 5 would be very useful already.**

To address this comment, the following adjustments were made to the text:

Line 224:
*The scientific interpretation of each factor was based on a combination of (i) the characteristics of the factor profile (i.e., "mass spectrum"), as referenced to a worldwide database of AMS spectra and PMF analyses (Ulbrich et al., 2009c; Ulbrich et al., 2009b, 2009a), and (ii) the temporal correlations between the factor loading and other co-located measurements (Figure 5).*

Line 233:
*Figure 5 shows that the correlations of factor loadings with external measurements of gas- and particle-phase species support this interpretation, as detailed in the following discussion for each factor.*

**My other comment is related to Figure 4: I agree with the other referee that it contains a lot of information and is hard to digest, but I think the authors are giving very good guidance through it in the text. I am not fully convinced that panel (c) in Figure 4 is really necessary. I am not sure how much it adds to the information already presented in the other panels and in Figure 5. I understand if the authors want to keep it as it is also.**

We thank the referee for this comment. We agree that Figure 4c is not essential for the understanding of the paper. It does however add value to the presentation, and we therefore chose to keep it, specifically for the following two reasons: (A) completeness of the presentation of results from the PMF analysis, which consist of factor profiles (Figure 4a) and loading time series (Figure 4c), and (B) visualization of some important correlations, which in Figure 5 are represented only as numbers.

[revised manuscript text omitted]

(a)

[Figure]

(b)

*Figure 1*

[Figure]

*Figure 2*

[Figure]

*Figure 3*

[Figure]

Figure 4

[Figure]

Figure 5

[Figure]

Figure 6

[Figure]

*Figure 7*

[Figure]

Figure 8

[Figure]

[Figure]

*Figure 9*